# Ocean variability beneath Thwaites Eastern Ice Shelf driven by the Pine Island Bay Gyre strength

Tiago S. Dotto [1] ✉, Karen J. Heywood [1], Rob A. Hall [1], Ted A. Scambos [2], Yixi Zheng [1], Yoshihiro Nakayama [3], Shuntaro Hyogo[4], Tasha Snow[5], Anna K. Wåhlin [6], Christian Wild[7], Martin Truffer[8], Atsuhiro Muto[9], Karen E. Alley[10], Lars Boehme [11], Guilherme A. Bortolotto [11], Scott W. Tyler [12] & Erin Pettit[7]

West Antarctic ice-shelf thinning is primarily caused by ocean-driven basal melting. Here we assess ocean variability below Thwaites Eastern Ice Shelf (TEIS) and reveal the importance of local ocean circulation and sea-ice. Measurements obtained from two sub-ice-shelf moorings, spanning January 2020 to March 2021, show warming of the ice-shelf cavity and an increase in meltwater fraction of the upper sub-ice layer. Combined with ocean modelling results, our observations suggest that meltwater from Pine Island Ice Shelf feeds into the TEIS cavity, adding to horizontal heat transport there. We propose that a weakening of the Pine Island Bay gyre caused by prolonged sea-ice cover from April 2020 to March 2021 allowed meltwater-enriched waters to enter the TEIS cavity, which increased the temperature of the upper layer. Our study highlights the sensitivity of ocean circulation beneath ice shelves to local atmosphere-sea-ice-ocean forcing in neighbouring open oceans.

Thwaites Glacier is among the fastest flowing marine-terminating glaciers in West Antarctica, and its grounding-line-retreat rate has more than doubled in the past two decades[1,2]. Between 1979 and 2017, Thwaites Glacier contributed 1.8 mm to global sea level, with nearly half of that contribution in the most recent decade, making Thwaites the largest net contributor to sea-level rise among Antarctic glaciers[3]. The eastern glacier front is buttressed by the Thwaites Eastern Ice Shelf (TEIS), which has thinned and weakened significantly in recent decades due to basal melting, mostly concentrated at the grounding zone[2,4,5]. High thinning rates at the grounding zone (~5 m yr$^{-1}$ between 2003 and

2020; ref. 4.), and rapid grounding-line retreat (0.3–0.8 km yr$^{-1}$ during 1992–2017; ref. 2, and up to 1.6 km yr$^{-1}$ since 2017; ref. 5) support that Thwaites Glacier and TEIS are in the early stages of rapid mass loss and retreat that may persist for decades to centuries[6]. However, the pace of such retreat is uncertain and dependent on future trends in climate and ocean conditions[7–9].

The retreat of Thwaites Glacier and TEIS is sustained by ocean circulation in the Amundsen Sea[10,11]. Increasing westerly wind speed and a southward shift of the boundary between westerly and easterly winds has raised the level of modified Circumpolar Deep Water

[1]Centre for Ocean and Atmospheric Sciences, School of Environmental Sciences, University of East Anglia, Norwich, UK. [2]Earth Science and Observation Center, Cooperative Institute for Research in Environmental Sciences, University of Colorado Boulder, Boulder, CO, USA. [3]Institute of Low Temperature Science, Hokkaido University, Sapporo, Japan. [4]Graduate School of Environmental Science, Hokkaido University, Sapporo, Japan. [5]Department of Geophysics, Colorado School of Mines, Golden, CO, USA. [6]Department of Marine Sciences, University of Gothenburg, Gothenburg, Sweden. [7]College of Earth, Ocean, and Atmospheric Sciences, Oregon State University, Corvallis, OR, USA. [8]Geophysical Institute and Department of Physics, University of Alaska Fairbanks, Fairbanks, AL, USA. [9]Department of Earth and Environmental Science, Temple University, Philadelphia, PA, USA. [10]Centre for Earth Observation Science, University of Manitoba, Winnipeg, MB, Canada. [11]Scottish Oceans Institute, University of St Andrews, St Andrews, UK. [12]Department of Geological Sciences and Engineering, University of Nevada, Reno, NV, USA. ✉e-mail: t.segabinazzi-dotto@uea.ac.uk

(mCDW) above the deeper bathymetry areas of the shelf break, allowing it to spread over the continental shelf[12]. Once on the continental shelf, mCDW is topographically steered towards the ice-shelf cavities through deep troughs (Fig. 1a). mCDW then floods the lower part of the sub-ice-shelf ocean cavities and induces widespread basal melting along the glaciated West Antarctica[13–15]. Recent observations[15,16] and ocean simulations[11] suggest that the eastern side of TEIS is fed by mCDW that has previously circulated near Pine Island Bay (PIB). An autonomous-underwater-vehicle survey below the ice-shelf front near the western side of TEIS also showed the presence of mCDW from PIB at 800–1000 m depth, indicating that there is a deep connection below TEIS through which deep warm water flows westward[15]. The extent of this deep connection, the pathways of the intermediate and upper ocean circulation, the driving forces, and any temporal variability of the currents under TEIS, are still unknown.

A cyclonic gyre located in the centre of PIB (hereafter PIB gyre[17]) conveys mCDW towards the Pine Island Ice Shelf (PIIS) cavity[13], entraining and spreading meltwater from PIIS northwestwards[18]. Both the glacial meltwater component from sub-ice-shelf cavities and increasingly glacially modified mCDW are then exported westwards along the Amundsen Sea coast[17,19–21]. Although meltwater-enriched waters from PIIS are spread within the PIB region, it is not clear if they reach the TEIS cavity and what consequences that might have. Idealised studies suggest that the presence of lighter and fresher waters in ice-shelf cavities could increase basal melting by intensifying the

horizontal circulation just beneath the ice base[22]. However, the effect of the interaction of exported meltwater on adjacent ice shelves is not well understood.

Here we use in situ oceanographic observations to show that the TEIS cavity warmed between January 2020 and March 2021, and simultaneously the glacial-meltwater content increased in the upper layers. We show via observations and ocean modelling that PIIS is a substantial source of the meltwater found in the TEIS cavity. The interannual oceanographic variability observed beneath TEIS is proposed to be linked to the weakening and strengthening of the PIB gyre, caused by local variability in landfast sea-ice cover. A weaker gyre causes the isopycnals beneath the ice shelf to be uplifted during sea-ice-covered periods, which brings warm thermocline waters upward. When the sea-ice-covered periods are prolonged, warming is caused by higher meltwater content that has accumulated in the PIB gyre and enters the cavity in the upper layers. The variability of the PIB gyre intensity is driven by local atmosphere−sea-ice−ocean interaction and may also control ice-shelf-cavity temperature in other similar systems around Antarctica.

## Results
### Interannual variability in ocean conditions beneath TEIS
Two Automated Meteorology-Ice-Geophysics-Ocean System (AMIGOS) moorings were deployed through TEIS, recording temperature, salinity, pressure, and horizontal current velocity at two depths

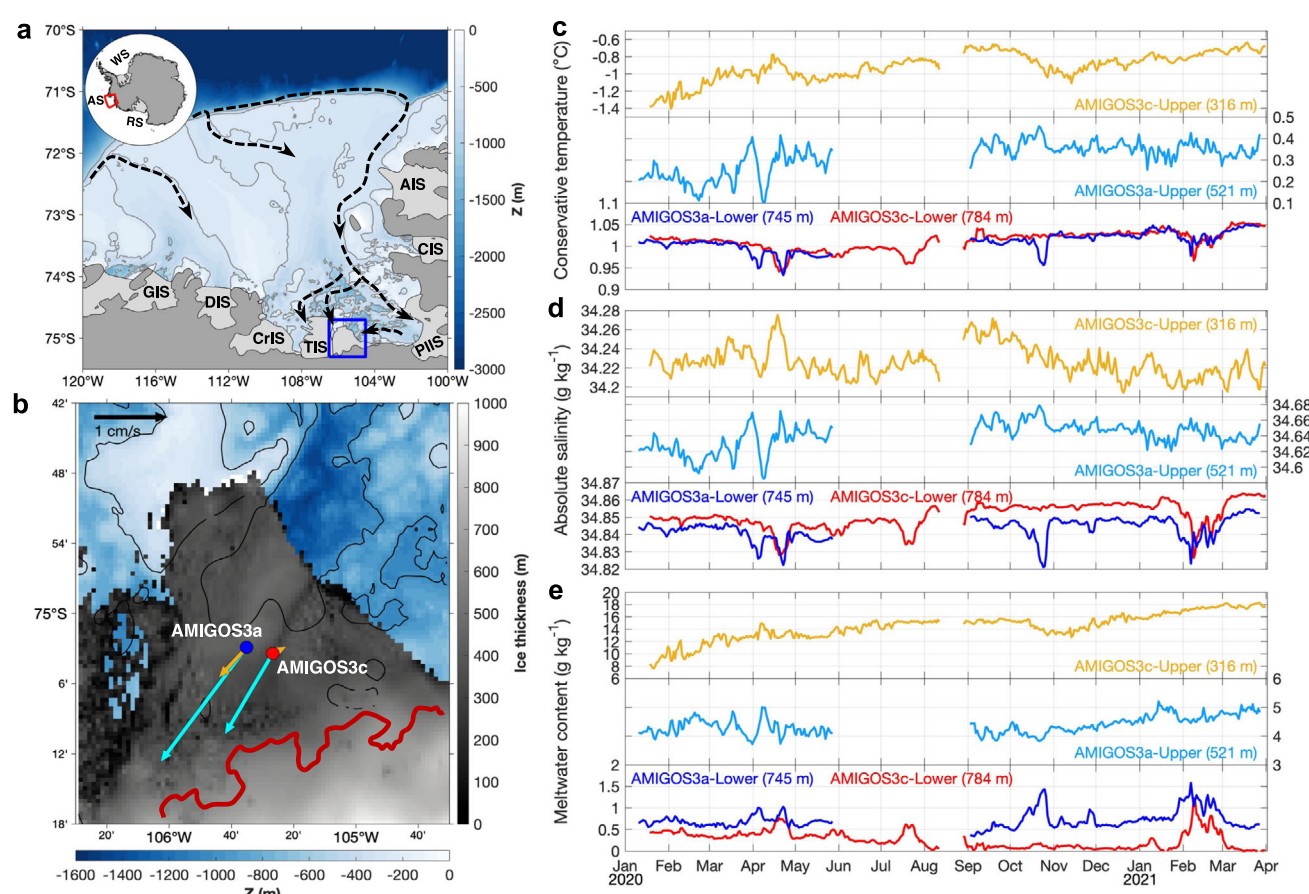

**Fig. 1 | Study region and hydrographic properties variability. a** Bathymetry of the continental shelf of the Amundsen Sea showing a schematic of the main pathways of modified Circumpolar Deep Water (mCDW; back dashed arrows) and ice shelves: Abbot Ice Shelf (AIS), Cosgrove Ice Shelf (CIS), Pine Island Ice Shelf (PIIS), Thwaites Ice Shelf (TIS), Crosson Ice Shelf (CrIS), Dotson Ice Shelf (DIS), and Getz Ice Shelf (GIS). Thwaites Eastern Ice Shelf (TEIS) is delimited by the blue rectangle. **b** TEIS study region showing the AMIGOS3a (blue dot) and AMIGOS3c

(red dot) sites. Ice thickness, bathymetry, and grounding line (dark red) from BedMachine Antarctic v2, ref. 37. Time-mean (January 2020 to March 2021) velocity vectors are shown for the upper sensors (cyan arrow) and deeper sensors (orange arrow) at the AMIGOS3a and AMIGOS3c sites. **c** Daily conservative temperature for the different sensors according to the legend. The mean depth of each sensor is shown in the legend. **d** Daily absolute salinity records. **e** Daily meltwater content estimated from conservative temperature and absolute salinity.

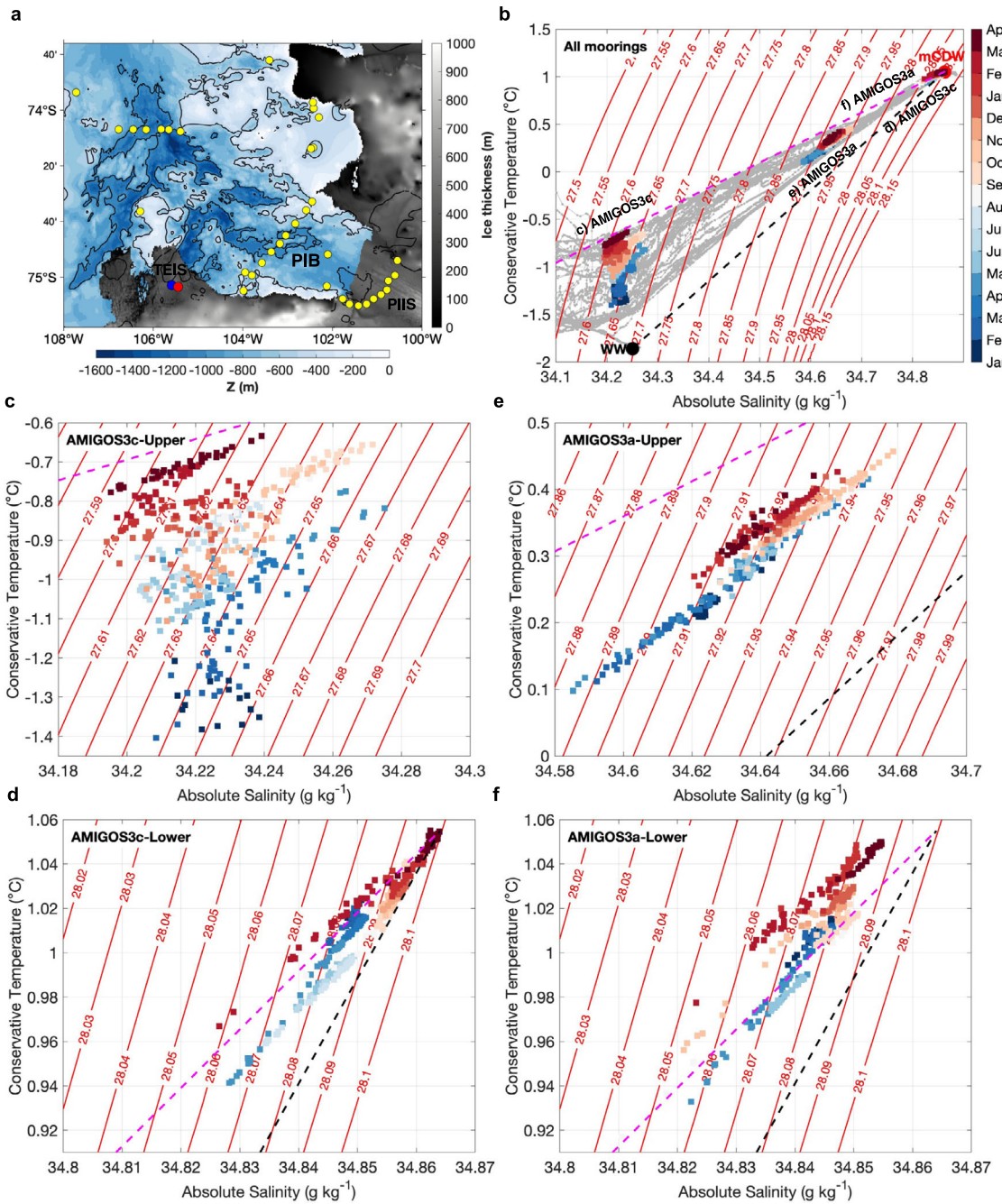

**Fig. 2 | Conservative Temperature-Absolute Salinity-Time diagrams. a** Location of Conductivity-Temperature-Depth (CTD) stations (yellow dots) in Pine Island Bay (PIB) collected in January and February 2020, and the AMIGOS3a (blue dot) and AMIGOS3c (red dot) sites. Thwaites Eastern Ice Shelf (TEIS) and Pine Island Ice Shelf (PIIS) are depicted. Ice thickness and bathymetry from BedMachine Antarctic v2, ref. 37. Note that BedMachine is not updated to the most recent calving front. **b** Conservative Temperature-Absolute Salinity diagram for the CTDs (grey dots) and AMIGOS colour-coded by the month that the measurement took place. Red lines are neutral density isopycnals. Magenta dashed is the Gade line[23] and black dashed is the mixing line between modified Circumpolar Deep Water (mCDW) and Winter Water (WW). Zoom-in for **c**, AMIGOS3c-Upper, **d** AMIGOS3c-Lower, **e** AMIGOS3a-Upper, and **f** AMIGOS3a-Lower.

between January 2020 and March 2021 (Methods; Fig. 1b). The first mooring was installed in a region of the ice shelf with relatively smooth surface topography (hereafter, AMIGOS3a; upper sensor at 521 m depth below sea level, and lower sensor at 745 m). The second mooring was installed beneath a basal channel (hereafter, AMIGOS3c; upper sensor at 316 m, and lower sensor at 784 m), at a distance ~4 km from the first. The in situ estimated ice-shelf drafts at the two sites were ~256 m and ~202 m, respectively (Supplementary Fig. 1).

Ocean characteristics indicative of mCDW were observed in the TEIS cavity at depths below ~650 m, as measured by the lower AMIGOS

sensors (Figs. 1c, d and 2a, b) and two borehole Conductivity-Temperature-Depth (CTD) profiles recorded before the deployment of the moorings (Supplementary Fig. 2). The layer below 650 m had little vertical variation, and the time-mean conservative temperature and absolute salinity were relatively constant, at -1.01 ± 0.02 ˚C and -34.85 ± 0.01 g kg⁻¹ (Fig. 1c, d), similar to the deep water found at the western side of TEIS[14]. A transitional layer between the warm and salty waters at deeper depths and cold and fresh waters at shallower depths was observed at the upper sensor on AMIGOS3a (~521 m; Supplementary Fig. 2), with time-mean conservative temperature and absolute

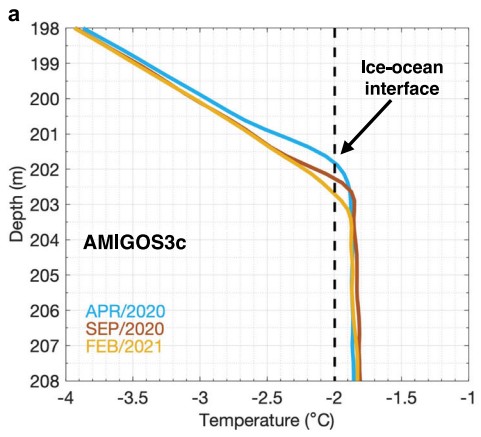
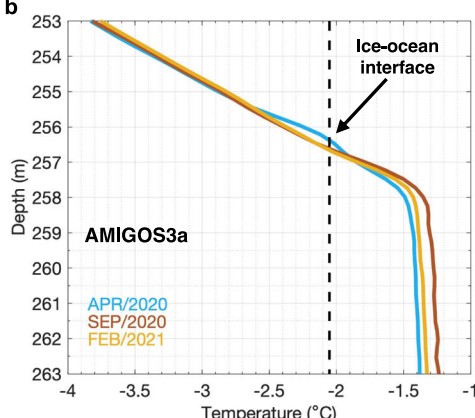

**Fig. 3 | Monthly averaged in situ temperature profiles beneath Thwaites Eastern Ice Shelf (TEIS) measured by fibre-optic cables. a** In situ temperature recorded at AMIGOS3c for April 2020 (light blue), September 2020 (red), and February 2021 (orange). **b** Same as **a**, but for AMIGOS3a. The vertical dashed lines are the in situ freezing temperature, assuming absolute salinities of 33.85 g kg⁻¹ for AMIGOS3c and 34.10 g kg⁻¹ for AMIGOS3a taken from the borehole profiles (Supplementary Fig. 2). Note that the y-axis limits are different between the two panels.

salinity at -0.31 ± 0.07 °C and -34.64 ± 0.02 g kg⁻¹ (Fig. 1c, d). The cold (-−0.91 ± 0.16 °C) and relatively fresh (-34.23 ± 0.02 g kg⁻¹) layer measured by the upper sensor of AMIGOS3c, at -100 m beneath the ice base, showed larger interannual variability than the deeper layers (Fig. 1c, d). The thermohaline variations at the two sites might be explained by vertical excursions (e.g., by internal waves or changes in baroclinic currents) or by horizontal advection of waters of differing properties. We combine the AMIGOS3c-Upper time series and the borehole CTD profiles to independently estimate possible vertical displacements of isotherms, isohalines and isopycnals that could account for the variations observed in the time series (Methods). While vertical isopycnal and isohaline displacements indicate possible movement of parcels of water up and down by tens of metres on daily to inter-seasonal timescales, the vertical isotherm displacements are not consistent, which they would have to be if the water mass property changes were entirely caused by vertical displacements. The vertical isotherm displacements show mostly an upward movement (Supplementary Fig. 3a), which is not consistent with the decrease in density observed in AMIGOS3c-Upper over time (Fig. 2c). Moreover, the reconstructed salinity associated with the vertical isopycnal displacement shows similar variability to the observations at AMIGOS3c-Upper (Supplementary Fig. 3b), whereas the reconstructed temperature does not show warming (Supplementary Fig. 3c). These results indicate that the excess heat observed at the upper AMIGOS must be caused by a different process other than simply vertical displacement of water masses. From the AMIGOS time series, temperature and salinity did not always increase together in the shallower layer (such as in January–March 2021), so vertical displacements of temperature, salinity and density cannot account for all of the temporal variability seen. For example, warming and freshening were observed occasionally (Fig. 1c, d), indicating that horizontal advective processes are also important in altering the water-mass properties there by bringing another water mass (such as freshwater input from ice melting) into these depths.

The upper layer of the TEIS cavity warmed between January 2020 and March 2021 (Figs. 1c and 2). The temperature increased by -1 °C in the upper layer at AMIGOS3c (Fig. 2c), while salinity increased to -34.26 g kg⁻¹ in September/October 2020, then decreased by -0.04 g kg⁻¹ (Fig. 1d). Higher temperature simultaneously with a decrease in salinity (e.g., after September/October 2020) indicates increased meltwater fractions[18,20,23,24] (Fig. 2b), because meltwater tends to be incorporated into the warm deep water that caused basal melting. Glacial-meltwater content was calculated from temperature and salinity[23,24] (Methods). While the meltwater content in the deepest layer remained virtually constant over the 14 months, it nearly doubled in the upper layer, from -10 g kg⁻¹ in February 2020 to -18 g kg⁻¹ in March 2021 at the upper sensor of the AMIGOS3c (Fig. 1e). In situ localized estimates of basal melting both within and outside of the basal channel from fibre-optic cables[25] (Methods) coupled to the AMIGOS moorings show small changes in the ice-ocean interface depth (Fig. 3), despite the rise in meltwater content (Figs. 1e and 2b–d). The monthly averaged profiles of April 2020, September 2020, and February 2021 show that the ice-ocean interface remains virtually steady at -202 m at AMIGOS3c (Fig. 3a) and -256.5 m at AMIGOS3a (Fig. 3b), assuming that the in situ freezing point is a good indicator of the interface. Historical data have shown that these sites are located within an area of low melting[4], which confirms the small changes in the estimated depth of the ice-ocean interface. While these point measurements are not necessarily indicative of the distributed melt rates, the lack of evidence of local melt does suggest that the rise in meltwater observed at these locations might have been advected into the TEIS cavity from elsewhere.

The flow beneath TEIS was southwestward at most depths (Fig. 1b). The speed in the upper and intermediate layers (generally not exceeding 10 cm s⁻¹) was higher than in the deeper layers (generally not exceeding 5 cm s⁻¹) and the flow was mostly oriented in a southwest-northeast direction (Fig. 4a–d). The observed current speeds are comparable to autonomous-vehicle observations beneath PIIS[13] and model simulations beneath PIIS and TEIS[11]. Conservation of potential vorticity can prevent ocean currents from moving across the calving front of ice shelves into their cavities[26,27]. The fact that a southwestward flow is observed in TEIS indicates that either the currents are mainly baroclinic, or that there are contours of constant water-column thickness that it can follow−i.e., steered by the isobaths of ice draft or by the seabed[27]. While it is interesting that the flow follows the direction of the basal channel in both moorings (Fig. 4a–d), our observations cannot provide further evidence whether the channel steers the flow into the TEIS cavity, or whether the flow influences the channel orientation. If the inflow was mainly barotropic and it was steered by the basal channel, the southwestward flow at AMIGOS3c should be significantly larger than that at AMIGOS3a. Instead, the time-mean flow speed at AMIGOS3a is larger than that at AMIGOS3c (Fig. 1b). This suggests that the inflow is predominantly baroclinic, and the ice front might present no topographic barrier to the inflow.

The southwestward flow direction below the ice shelf (Figs. 1b and 4a–d) indicates that thermocline water is imported from PIB. This is supported by previous results that showed that the deep and intermediate waters moved from PIB below TEIS towards the region near

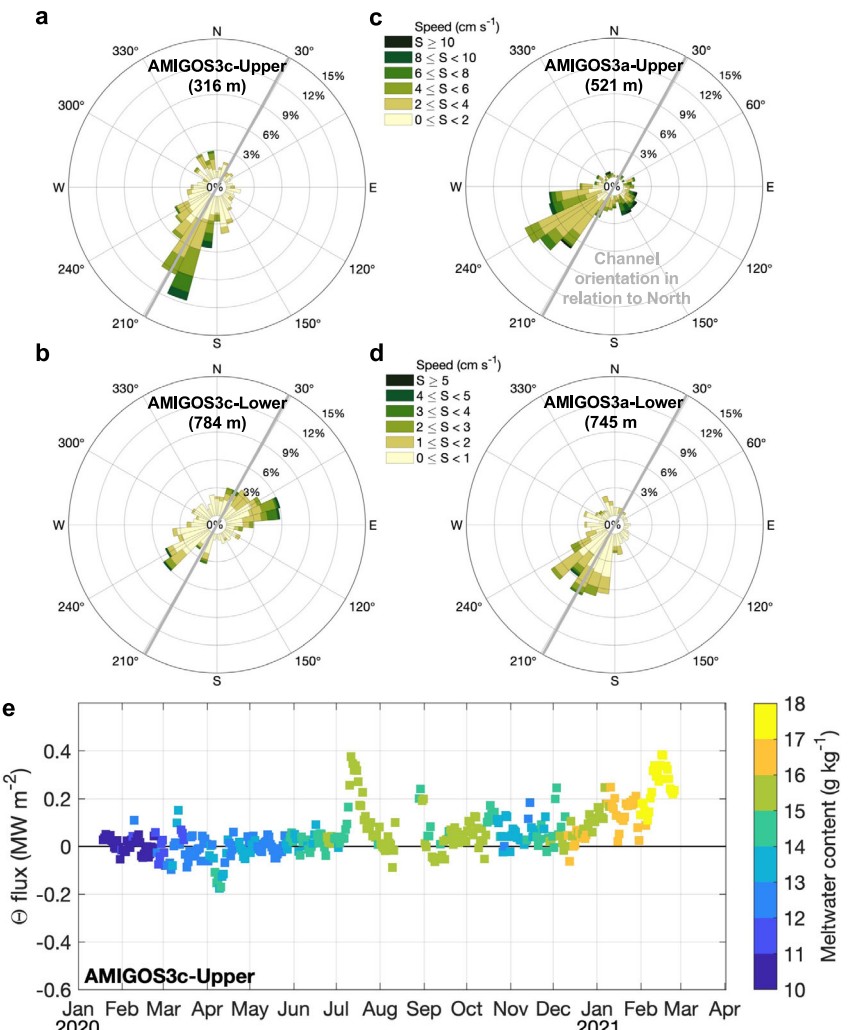

**Fig. 4 | Ocean current variability and temperature flux.** Current-rose by occurrence for different speeds and directions for **a** AMIGOS3c-Upper, **b** AMIGOS3c-Lower, **c** AMIGOS3a-Upper, and **d** AMIGOS3a-Lower. The mean depth of each sensor is shown in parentheses. Note that the speed scale varies between Upper (**a**, **c**) and Lower (**b**, **d**) sensors. The thick grey line depicts the sub-ice shelf channel oriented to North. **e** Conservative temperature (Θ) flux from the AMIGOS3c-Upper coloured by meltwater content.

and underneath the northern opening between TEIS and Thwaites Ice Tongue[15]. Further evidence comes from the water-mass properties (Supplementary Fig. 4). It is striking to observe that the thermohaline properties at the AMIGOS sites are consistent with the water-mass properties collected from the west and north of Thwaites Ice Shelf by autonomous-underwater-vehicle and ship-based Conductivity-Temperature-Depth profiler in 2019[15] (Supplementary Fig. 4b–e). This suggests that such a connection between PIB and the west of Thwaites Ice Shelf occur year-round because the deep properties overlap on the same temperature-salinity space from the measurements of 2019 (Supplementary Fig. 4c–e). We also suggest that changes in water masses beneath TEIS likely occur at all depths and not only at the deeper layers[15] (e.g., the water masses at the AMIGOS3c-Upper share similar temperature-salinity space with the measurements of 2019 in the west and north openings; Supplementary Fig. 4b). These results suggest that the water masses access the central cavity below TEIS from the east, indicating that the connection for deep water[15] is part of a large-scale westward flow of water masses including intermediate and upper depths.

A layer of light, meltwater-enriched waters could contribute indirectly to accelerating the under-ice circulation by creating horizontal density gradients[22,28], with potential for enhancing local basal melting[29]. By combining temperature and velocity measurements, we calculated temperature flux (Methods), a proxy for the heat flux, delivered into the TEIS cavity by the inflow. The along-channel temperature flux in AMIGOS3c-Upper showed three peaks: July 2020, late August 2020, and February 2021, of up to ~0.2–0.4 MW m$^{-2}$, which were associated with the lighter waters having higher meltwater content (Fig. 4e and Supplementary Fig. 5). Higher heat flux could increase basal-channel erosion, causing ice fracture and weakening of the ice shelf (e.g., ref. 30). However, the occasional peaks in temperature flux at the AMIGOS3c site (Fig. 4e) either did not reach the base of the ice shelf, and/or were not enough to cause a strong melting event (e.g., Fig. 3), which further highlights the complexity of the TEIS melting.

**Meltwater pathways between PIIS and TEIS**

The observations presented here indicate that the light, meltwater-enriched water found in PIB[20] is advected into the TEIS cavity (Figs. 1, 2, and 4). We use a high-resolution model[11] to test whether it is feasible for the meltwater exported from PIIS to access the TEIS cavity. We release particles from the western side of the PIIS cavity (Methods), the region where most meltwater is exported from beneath PIIS[17,20], and track them forward in time. The particles initially follow the basal channels carved into PIIS and then flow westward with the coastal

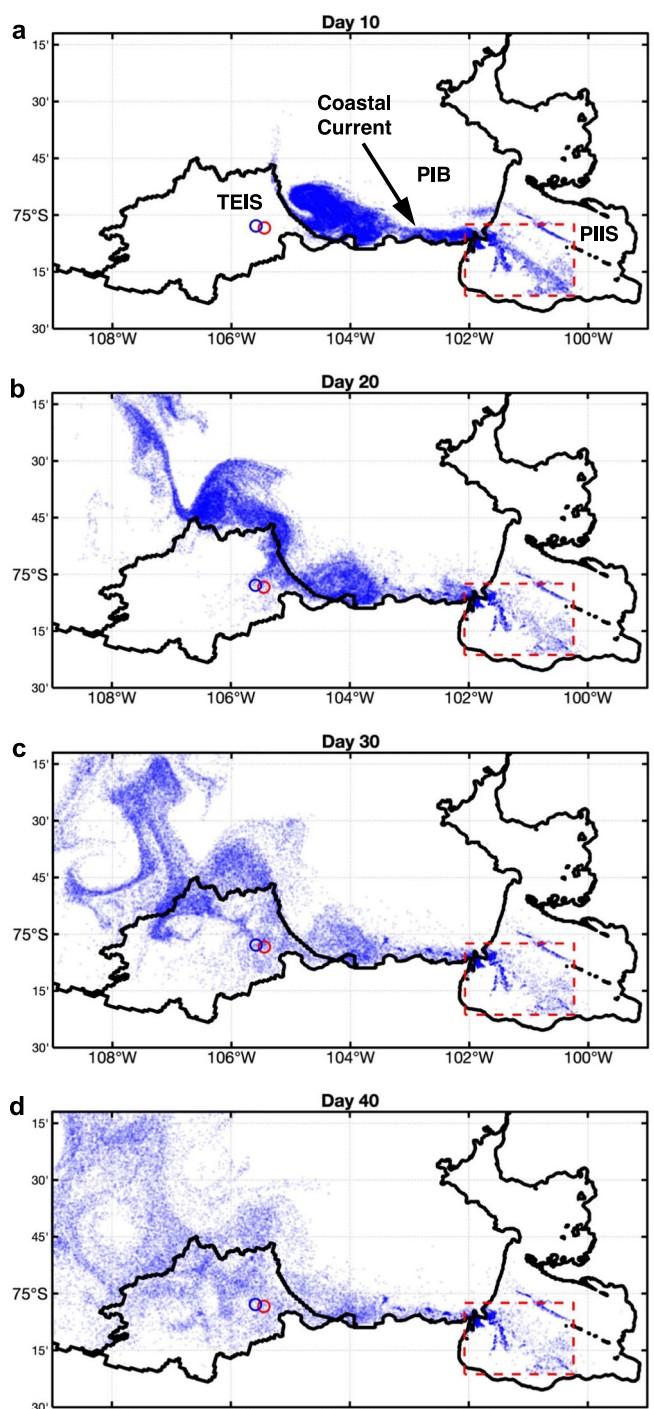

**Fig. 5 | Trajectories of simulated particles released in the Pine Island Ice Shelf (PIIS) cavity (red rectangle).** A total of 200645 particles were released beneath PIIS in the red rectangle in an offline simulation. For illustration purposes, the particles were restricted to meltwater content of 10–25 g kg⁻¹ and depths 250–400 m, based on day 2 of the simulation (when particles leave the cavity), which reduces the total number of particles to 42521. The snapshots show particle location for days **a** 10, **b** 20, **c** 30, and **d** 40 of the simulation. Open blue and red circles depict AMIGOS3a and AMIGOS3c sites, respectively. Thwaites Eastern Ice Shelf (TEIS) and Pine Island Bay (PIB) are depicted.

current (Fig. 5). The leading particles reach the TEIS calving front after 10 days (Fig. 5a). By days 12 and 15, particles reach the AMIGOS sites (Supplementary Movie 1), although their access is not necessarily concentrated at the basal channel since the model ice shelf does not properly represent this feature. From day 20, particles are present

throughout the TEIS cavity (Fig. 5b). The whole Thwaites Ice Shelf cavity, above 400 m, is occupied by particles between days 30 and 40 (Fig. 5c, d). These results support the hypothesis that outflow from the PIIS cavity feeds the TEIS cavity on a relatively short timescale.

Using wintertime profiles from 2020 obtained by seal tags (Fig. 6a), we compare the properties of the suggested source in PIB with the observed water underneath TEIS. The temperature-salinity relationship and meltwater content calculated from seal-tag and the mooring data show that in June-July 2020, the properties of the PIB water at 200–300 m depth were similar to what was observed at the mooring site (~316 m). These data points overlap in temperature-salinity space, suggesting a common origin (Fig. 5c). The meltwater content was comparatively high outside TEIS at 200–300 m depth, ~12–14 g kg⁻¹ (Fig. 5b), and within its cavity at the AMIGOS moorings. The meltwater concentration in June-July 2020 was higher than observed in previous years for similar depths[18]. Based on the seal-tag and model results presented here, we suggest that PIIS is a significant source of meltwater within the TEIS cavity, and of the meltwater exported at the western side of the Thwaites Ice Shelf[15]. Therefore, we argue that both ice shelves need to be assessed together to better reveal the drivers of variable ocean conditions beneath Thwaites Ice Shelf and to understand and forecast its future evolution.

## Driving forces for the flow under TEIS

We now investigate possible drivers for the thermohaline variability observed beneath TEIS (Fig. 2). The seawater properties at the upper sensor of AMIGOS3c, which is in the upper layer, were distinct in different seasons (Fig. 1c–e). January to March 2020 exhibited low temperature, relatively high salinity and density, and low meltwater content. Conversely, August to October 2020 exhibited high temperature, salinity, density, and meltwater content (although meltwater content was high, it did not increase during these months; Fig. 1e). January to March 2021 exhibited high temperature and meltwater content but low salinity and density. Here, we hypothesize that the variability[18,31] of the ~50 km diameter cyclonic PIB gyre[17] (Fig. 7a) controls the properties of the water flowing into the Thwaites cavity, as it depresses or raises the isopycnals near the TEIS front.

Theories suggest that when the ocean is sea-ice free and/or the sea-ice is mobile, momentum transfer from the wind into the ocean is higher (e.g., ref. 32), which can intensify the PIB gyre[33]. This is further confirmed by a model simulation which shows steeper isopycnals at the gyre's boundary in February–March 2020 (i.e., sea-ice-free conditions; Fig. 7b). A stronger gyre depresses the isopycnals at its boundary and permits cold surface water to flow into the upper layer beneath TEIS, as it did in January and March 2020 (Figs. 1c–e and 8a). Conversely, landfast sea-ice cover or high concentration of less-mobile sea-ice in PIB reduces the surface momentum transfer into the ocean, weakening the gyre[33]. Our model confirms the weakening of the PIB gyre in the sea-ice-covered period of August-September 2020 (Fig. 7b). The model also shows that a weaker gyre shoals the isopycnals beneath TEIS in that period. The shoaling of the isopycnals in those months is consistent with the upward expansion of the thermocline layer (Supplementary Figs. 2a and 3a), which brings warmer and saltier waters in the upper layer, as observed between August and October 2020 in AMIGOS3c-Upper (Figs. 1c, d and 8b). The increase in both temperature and salinity does not increase the meltwater further (Fig. 1e – note that the meltwater content stays virtually the same during those months, below 16 g kg⁻¹). This seasonal change in the depth of the isopycnals beneath TEIS in the model is in the order of tens of metres (Fig. 7c), which coincides with the estimated isopycnal vertical displacements (Supplementary Fig. 3a). Seal-tag hydrography from 2020 also supports these changes in the gyre strength: neutral density at 300 m depth was higher in the PIB centre in austral summer than in winter, consistent with a rise of the gyre's dome and stronger circulation in summer (Fig. 7d).

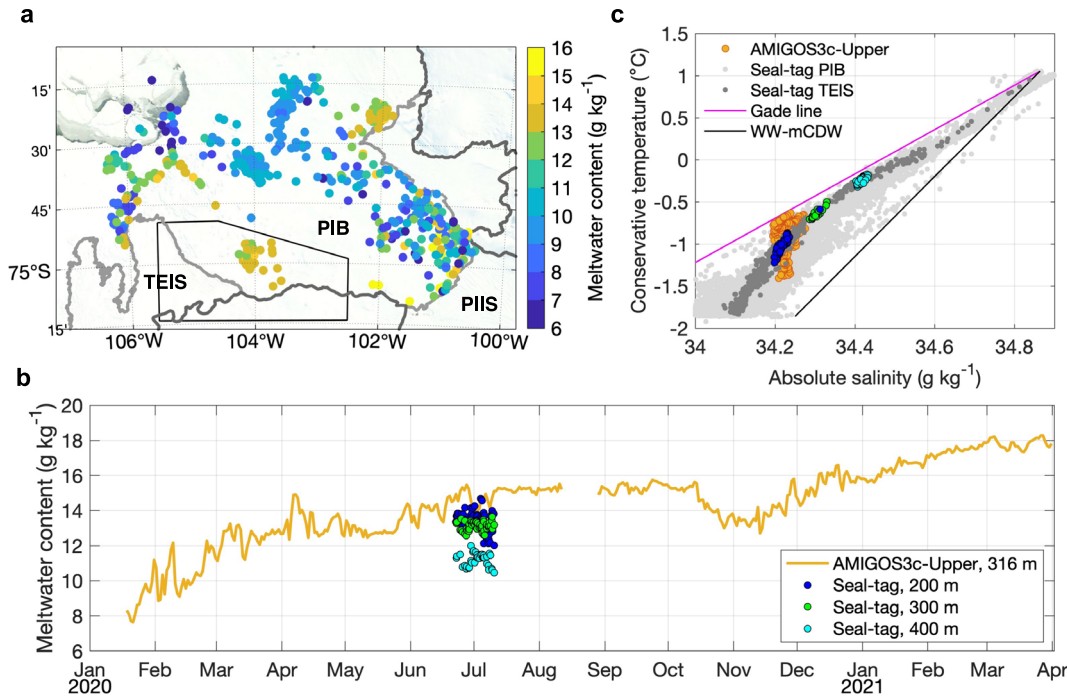

**Fig. 6 | Meltwater content at Pine Island Bay (PIB) from seal-tag data.**
**a** Meltwater content at 200 m depth derived from seal-tag data collected between May and October of 2020. The black polygon shows the area used to select the seal-tag data near Thwaites Eastern Ice Shelf (TEIS). Terra MODIS optical imagery for 10 September 2020 is shown. Grey contours show land (dark) and ice shelves (light). Pine Island Ice Shelf (PIIS) is depicted. **b** Time series of meltwater for AMIGOS3c-Upper (orange, same as in Fig. 1e) and for the seal-tag data at 200 m (blue), 300 m

(green), and 400 m (cyan) within the black polygon in "**a**". **c** Conservative temperature-absolute salinity diagram showing the hydrographic properties for AMIGOS3c-Upper (orange) and the seal-tag data (grey). Light grey represents all seal-tag data within PIB. Dark grey represents the seal-tag within the black polygon of "**a**", at 200 m (blue), 300 m (green), and 400 m (cyan). Magenta line depicts the Gade line[23] and black line represents the mixing line of and Winter Water (WW) and modified Circumpolar Deep Water (mCDW).

January to March 2021 was marked by a persistent landfast sea-ice covering PIB (contrarily to January to March 2020), which we suggest prolonged the weaker PIB gyre period (Fig. 9a). The model shows that in December 2020 (i.e., the end of the simulation) the PIB gyre was indeed weak, as suggested by the slightly flattened isopycnals (Fig. 7b). However, a weaker PIB gyre did not lead to the shoaling of the isopycnals beneath TEIS as it did in August-September 2020. In fact, the isopycnals in December 2020 deepened further beneath TEIS (Fig. 7b). The deepening of the isopycnals is consistent with a lightening of the upper layer, in agreement with the observations (Fig. 2c). A likely explanation for the lower density levels in the upper layer in that period is that the weakening of the PIB gyre might export less meltwater from PIB, accumulating it in the bay. This process is opposite to what has been observed in the Beaufort Gyre, Arctic, where the strengthening of the gyre leads to the accumulation of meltwater[34]. The higher concentration of meltwater-enriched waters within PIB may decrease the density of the upper layer, as meltwater is lighter than the ambient water because it is warmer and fresher. The deepening of the isopycnals at the TEIS front, as suggested by the model (Fig. 7b), thus allows the access of a higher volume of (shallower) meltwater-enriched waters into its cavity. This explains the warm (Fig. 1c), light (Fig. 2c), and high meltwater content (Fig. 1e) water observed in the AMIGOS3c-Upper in January to March 2021.

In summary, the strength of the PIB gyre controls the heaving of the isopycnals near and beneath TEIS, as shown by the ocean model (Fig. 7b), which allows a higher or lower volume of upper water to access the TEIS cavity. In austral winter (Fig. 8b), the weakening of the gyre uplift the isopycnals beneath TEIS and bring slightly warmer and saltier waters to shallower depths (Fig. 1c). Under prolonged periods of sea-ice cover (Fig. 8c), the weaker gyre might export less meltwater-enriched waters from PIB. This decreases the density at the upper

layers at PIB, which leads to a deepening of the isopycnals near TEIS front and facilitates the inflow of meltwater-enriched waters into its cavity.

## Discussion

Our observations of ocean conditions in the TEIS cavity showed a general warming of the water column between January 2020 and March 2021 (Figs. 1c and 2). In the upper layers, this warming was associated with an increase in the meltwater content entering the TEIS cavity (Figs. 1e and 2c). Our analysis suggests that a substantial fraction of the meltwater found within the TEIS cavity originates beneath PIIS (Fig. 5). We propose that the interannual thermohaline variability observed beneath TEIS is associated with the changes in the strength of the PIB gyre, caused by local variability in landfast sea-ice cover (Fig. 8). The weaker gyre during sea-ice-covered periods (e.g., August to October 2020) uplifts the isopycnals beneath TEIS, which brings thermocline warm water upwards (Fig. 8b). However, when the sea-ice-covered period is prolonged (e.g., January to March 2021), warming beneath TEIS is caused by higher meltwater content entering the cavity in the upper layers (Fig. 8c). The weaker gyre reduces the export of meltwater from PIB, which is then accumulated in the area, and reduces the density field in the upper ocean. Since the PIIS cavity is hydrographically similar to the TEIS cavity, we speculate that the weakening of the PIB gyre might lead to a warming and a potentially higher basal-melt rate in the PIIS cavity. These mechanisms might explain the higher meltwater content in both PIB and TEIS in 2020–2021 (Fig. 5b), and the warming of TEIS cavity over time (Fig. 1c). Thus, the PIB gyre might ultimately affect the long-term evolution of the Thwaites Ice Shelf.

Between 2002 and 2021, three austral summers (i.e., 2004, 2005, and 2018) presented comparable sea-ice concentrations to

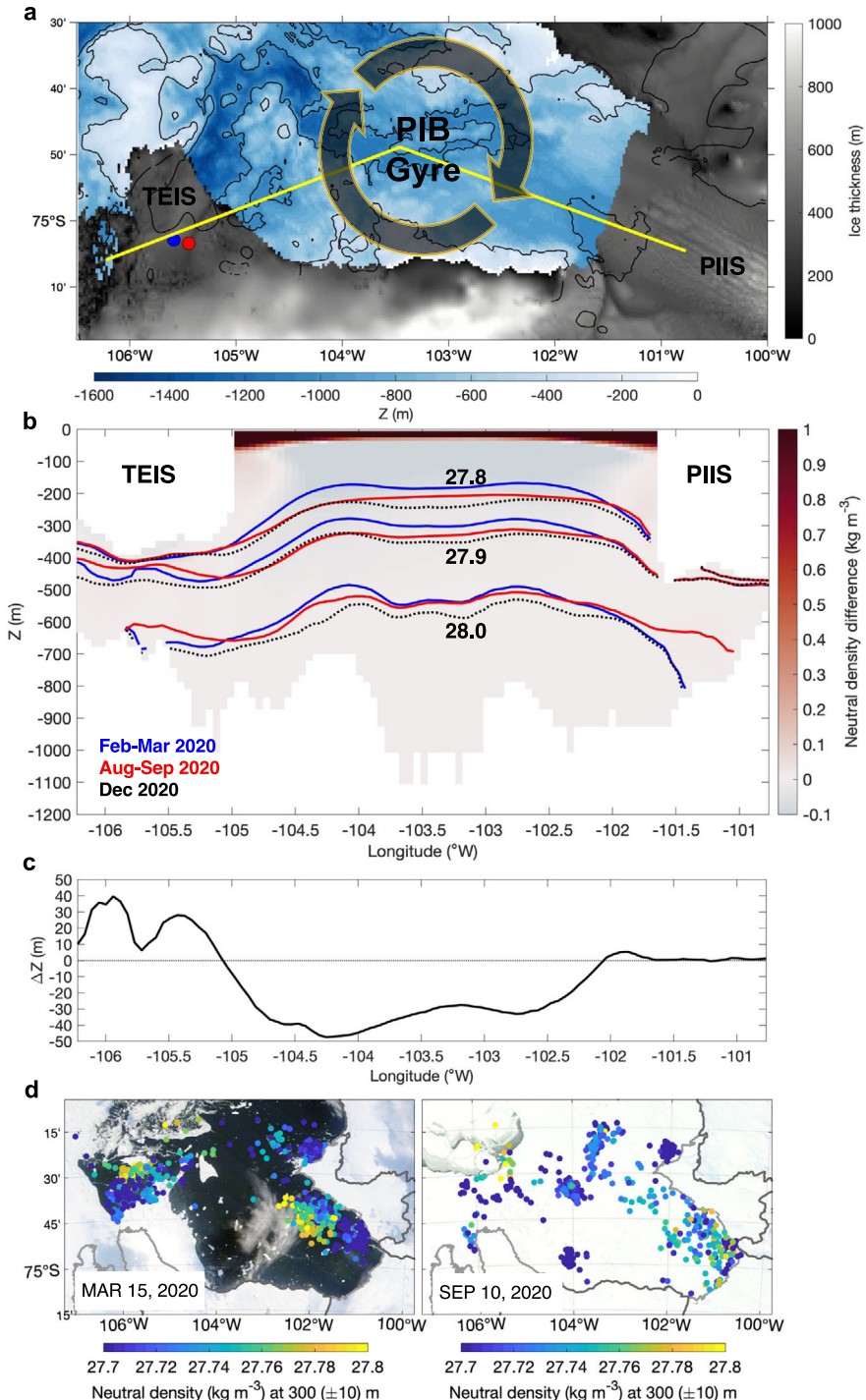

**Fig. 7 | Regional ocean simulation showing the variability of the Pine Island Bay (PIB) gyre density structure. a** Location of the section used in panel b and the AMIGOS3a (blue) and AMIGOS3c (red) moorings. Ice thickness and bathymetry from BedMachine Antarctic v2, ref. 37. Thwaites Eastern Ice Shelf (TEIS) and Pine Island Ice Shelf (PIIS) are identified in the map. Note that BedMachine is not updated to the most recent calving front. A schematic location of the PIB gyre is shown by the arrows. **b** Vertical section showing the difference in neutral density between August-September 2020 (red lines) and February–March 2020 (blue lines) to illustrate the density variation in the sea-ice-covered and sea-ice-free periods, respectively. During a sea-ice-covered period, the isopycnals deepen at the centre of the PIB gyre and shallow beneath TEIS, representative of spin-down of the gyre in winter. The isopycnals of December 2020 (prolonged sea-ice-cover conditions) are shown in black dashed lines. **c** Depth difference (ΔZ in m) of the isopycnal of 27.9 kg m$^{-3}$ between August–September 2020 and February–March 2020. **d** Neutral density calculated at the 300 m depth from seal-tag data for February-April 2020 (summer, left) and May–October 2020 (winter, right). Terra MODIS optical imagery for 15 March 2020 exemplifies a sea-ice-free condition and 10 September 2020 a sea-ice-covered period. Grey contours represent land (dark) and ice shelves (light).

January–March 2021 (Fig. 9b). Two of these years fall within the cold decadal period of early-2000s described for West Antarctica[13,14]. Our hypothesis implies, somewhat counter-intuitively, that these ice-shelf cavities could warm slightly more during cold ocean periods if that condition supports landfast sea-ice cover in PIB and adjacent to TEIS, by ultimately allowing more meltwater to enter the ice-shelf cavities, compared with sea-ice-free periods (Fig. 2c). However, there will be feedbacks upstream

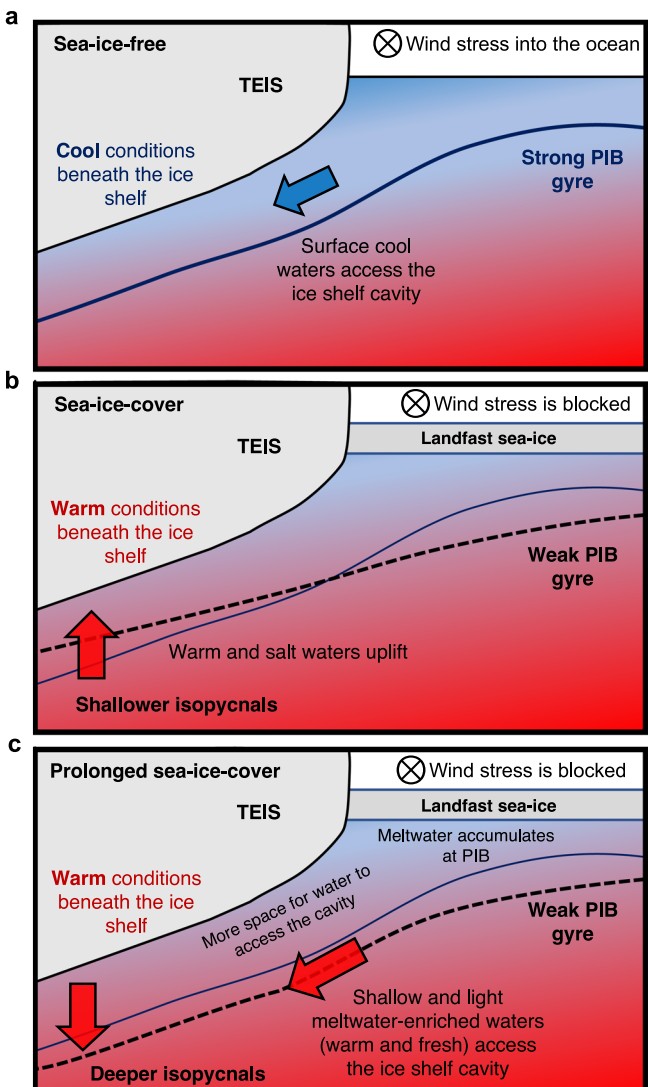

**Fig. 8 | Schematic of the processes identified. a** During sea-ice-free conditions, the Thwaites Eastern Ice Shelf (TEIS) cavity is filled with shallow cold waters due to the strengthening of the Pine Island Bay (PIB) gyre, as depicted by the steepening of the isopycnal (solid line). The density field is shallow in the middle of the gyre and deepens to beneath the TEIS front. **b** During sea-ice-covered periods, the wind-stress is damped and the isopycnals relax, which characterises a weakening of the gyre (dashed line). Under the TEIS, the isopycnals uplift, which brings warm and salt waters close to the ice base, creating a warm condition beneath the ice shelf. **c** During prolonged sea-ice-covered periods, the isopycnals flatten (yellow dashed line), the gyre weaken, and meltwater is accumulated in the PIB area, which reduces the density field. The isopycnals near and beneath TEIS deepen, which open space for higher volume of shallow and light meltwater-enriched waters to flood the upper layers of the ice shelf cavity, leading to a warm condition. In all panels the isopycnal of sea-ice-free conditions is depicted for reference. The colour illustrates the temperature field, with red (blue) colour representing warm (cold) waters.

that may alter the concentration of meltwater in adjacent ice shelves.

Our hypothesis relies on the fact that sea-ice coverage considerably dampens the wind stress over the ocean surface. Polar ocean gyres tend to weaken during periods of extensive sea-ice coverage and less mobile sea-ice (e.g., refs. 32–36.). Moreover, a gyre not only weakens under the presence of high concentrations of landfast-ice but may also reverse its direction from clockwise to anticlockwise depending on the strength and the angle between the wind direction and the sea-ice edge[33]. A change in the gyre direction could warm the

ice-shelf cavities even faster than the simple spin-down suggested in this work because it could lift the isopycnals further up beneath the ice shelf and bring deeper warm waters upwards.

Prolonged periods of weaker gyres lead to warmer conditions within ice-shelf cavities. Consequently, more glacial meltwater may be exported from one ice-shelf cavity to another. The meltwater imported from adjacent cavities suggests that ice shelves are coupled systems connected through the coastal circulation[15,19]. In this sense, what happens under one ice shelf greatly influences what happens under the ice shelves further downstream in the coastal current. Therefore, models should assess the meltwater pathways from adjacent ice shelves and their consequences at the ice-ocean boundary to better simulate the fate of the Antarctic ice shelves, at least in shelf regions such as the Amundsen Sea where rapidly thinning ice shelves releasing considerable freshwater into the ocean are geographically connected (e.g., ref. 21.).

Theoretical and modelling studies have suggested that the addition of meltwater can lead to increased temperature and stronger baroclinic circulation beneath an ice shelf[22,28]. This would be caused by altering the horizontal density gradients within the cavity with the potential for more basal melting through stronger vertical heat flux through mixing, posing a negative feedback system. If prolonged sea-ice-covered periods will be more common in the future, the Thwaites Ice Shelf cavity will be prone to warming, which makes it vulnerable to basal melting. However, there is no strong evidence that a high local basal melting is happening in our study sites, at least during the study period (e.g., Fig. 3). An alternative process could be that the meltwater-enriched water advected in from PIB is sufficiently buoyant to form a less dense upper layer below the ice base, increasing the stratification, stabilizing the water column, and suppressing the vertical heat flux, which could potentially reduce the local basal melting. This process may be different at the pinning points and at the grounding zone, where higher melting rates are historically observed[4,5]. A future study targeting these processes (and the peculiarities in different parts of the ice shelf) would need to deploy sensors close to the ice base and/or use bespoke boundary-layer models to properly address these complex interactions.

The atmosphere-sea-ice-ocean interactions discussed here are important because they can prolong warm periods beneath ice shelves by allowing warm and meltwater-enriched water to enter adjacent ice-shelf cavities. The key components are: (i) gyres that respond to wind forcing adjacent to the ice shelf, and (ii) prolonged periods of landfast-ice cover damping the momentum transfer from the wind and thus spinning-down the gyre. Gyres adjacent to ice shelves are relatively small features that are still being discovered through high-resolution surveys (e.g., the gyre recently identified to the west of the Thwaites Ice Tongue[33]). Gyres potentially existing in other regions around Antarctica (e.g., East Antarctica) may cause a greater number of ice shelves to be prone to intense basal melting associated with prolonged warm conditions.

## Methods
### Hydrographic measurements
The moorings are a component of a multi-sensor climate, ice, and ocean geophysical observation system (Automated Meteorology Ice-Geophysics-Ocean System, AMIGOS; see Scambos et al.[38] for a description of an earlier version). The AMIGOS were installed by drilling boreholes using hot water at two sites ~4.13 km apart on the Thwaites Eastern Ice Shelf: AMIGOS3c at the "basal channel site" (75.057˚S, 105.446˚W) and AMIGOS3a at the "plateau site" (75.048˚S, 105.586˚W) (Fig. 1b). Each AMIGOS mooring was equipped with two SeaBird MicroCAT SBE 37-IMP conductivity-temperature-pressure sensors and two Nortek Aquadopp single-point current metres (Supplementary Fig. 1b). At the channel site, the MicroCATs were installed at ~316 m and ~784 m, and the Aquadopps were installed ~2 m beneath

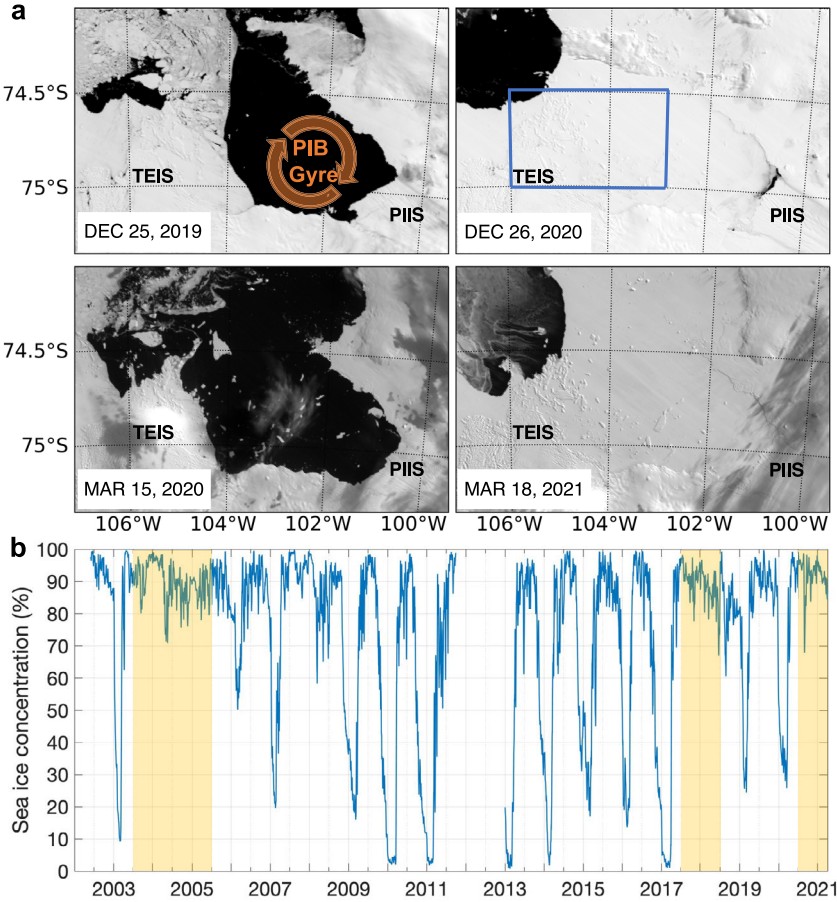

**Fig. 9 | Sea-ice concentration during the study period. a** Terra MODIS optical imagery shows the sea-ice coverage in Pine Island Bay (PIB) during two dates in early and late 2019–2020, and 2020–2021 austral summer seasons. Schematic of the cyclonic PIB gyre is shown in upper left panel. Blue polygon in upper right panel delimits the area where the sea-ice concentration was estimated for panel b.

Thwaites Eastern Ice Shelf (TEIS) and Pine Island Ice Shelf (PIIS) are depicted. **b** Daily sea-ice concentration from AMSR-E/AMSR2. The time series were smoothed by a 2-day running-mean window. Yellow shading highlights years of high summertime sea-ice concentration.

each MicroCAT. In the plateau site, the MicroCATs were installed at ~521 m and ~745 m, with Aquadopps installed ~2 m beneath each MicroCAT. The sensors sampled every hour from 10-January-2020 to 30-March-2021, and the data were transmitted via Iridium satellite telemetry. Data gaps (hours to weeks) were caused either by (i) failed telemetry (e.g., poor signal), (ii) bad weather (e.g., snow and/or high winds), or (iii) low power (e.g., the lack of enough sunlight during winter difficulted recharging the stations). These gaps were not interpolated. The velocity components measured by the Aquadopps were corrected for the magnetic declination, 50.07 ̊E. The hourly data were binned into daily means. Conservative temperature (Θ; ̊C) and absolute salinity ($S_A$; g kg$^{-1}$) were calculated from Thermodynamic Equations of Seawater-10 (ref. 39). Neutral density was calculated according to Jackett & McDougall[40].

Conductivity-temperature-depth (CTD) profiles were conducted in February and March 2020 on board RVIB Nathaniel B. Palmer using a SeaBird SBE911+. Conductivity was calibrated using a Guideline Autosal salinometer. The CTD profiles were 1-m binned. CTD profiles from boreholes were conducted on 31 December 2019 in the AMIGOS3a site, and on 13 January 2020 in the AMIGOS3c site using the MicroCAT installed at the deeper depth for each mooring. The profiles were interpolated at 1-m intervals and smoothed with 5-m running mean.

In situ temperature profiles beneath the TEIS were obtained from a Distributed Temperature Sensing (DTS) and armoured fibre-optic cable similar to that previously tested on the McMurdo Ice Shelf[25]

coupled to the AMIGOS. DTS utilizes a fibre-optic cable installed within the ice and through the ocean, and it uses Raman-backscattered photons to estimate the in situ temperature of the fibre[41]. The DTS interrogators (Silixa XT-DTS, Silixa LLC. Elstree, UK) used a spatial sampling of 25 cm and were set to sample 6 times per day, except for 1 sample per day in Austral winter to conserve power. Each DTS profile uses a 1-minute integration time to improve signal-to-noise ratio. Independent temperature measurements from the MicroCATs are employed to calibrate the backscatter signal[25,41]. Calibration consisted of a simple two-point offset and gain using the MicroCATs temperature. For AMIGOS3c, the temporal Root Mean Square Error, calculated as the difference between the MicroCAT temperature and the calibrated DTS temperature at the MicroCAT depths at near coincidence times, was ~0.04 ̊C at both 316 m and 784 m. Similar Root Mean Square Errors were found for AMIGOS3a. The DTS profiles were monthly averaged for illustration purposes.

Twelve and nine CTD-Satellite Relayed Data Loggers[42] were deployed in early 2019 and early 2020, respectively, in Weddell- and elephant-seals in the Amundsen Sea Embayment. All seal-tag profiles were compared with ship-based CTD measurements before deployment and the final received dataset was quality controlled using similar methods used by the Argo float community[43]. Spurious data were flagged and removed.

Temperature and salinity profiles shown in Supplementary Fig. 4 were collected in 2019 beneath and at the surroundings of Thwaites Ice Shelf. The data beneath Thwaites Ice Shelf were collected by a

Kongsberg Hugin Autonomous Underwater Vehicle equipped with two SeaBird SBE 19plus V2 systems. A Sea-Bird 911+ CTD was used to measure the data in the surroundings of the ice shelf. More information regarding data collection, post-processing and calibration is provided by Wåhlin et al.[15].

## Calculation of vertical water mass displacement

The vertical displacement of isotherm, isohaline, and isopycnals was calculated by combining the AMIGOS timeseries measurements and the vertical borehole CTD profiles (1-m binned and then 5-m running-meaned smoothed). For each time step in the moored time series at AMIGOS3c-Upper, we determine the depth in the borehole profile that has a similar potential density to the mooring value. The vertical displacement is then the difference in depth between the mooring and the depth in the profile that has the same density. We determine this separately for potential density, absolute salinity, and conservative temperature in order to have independent time series of vertical displacements for isopycnals, isohalines, and isotherms. Because the temperature profile is not monotonic (Supplementary Fig. 2a), early in the time series (up to 17th February 2020) there are two solutions for the isotherm displacement above and below the mooring depth; we show these two solutions as dots in Supplementary Fig. 3a. Later in the time series there is only one solution because the temperature is sufficiently high that it is only found below the mooring. We also estimated values of conservative temperature and absolute salinity associated with the vertical isopycnal displacements (Supplementary Fig. 3b, c, respectively).

## Calculation of meltwater content from hydrographic data

We calculated meltwater content following the composite-tracer method[24]. Tracers are $\Theta$ and $S_A$, and both are assumed to be conservative for all observations. We also assumed that the seawater in our study region is composed of the source water masses, mCDW, Winter Water, and glacial meltwater. Hence, the meltwater fraction can be derived from the observations following:

$$\varphi_{meltwater} = \frac{\Theta_{observed} - \Theta_{mCDW} - \frac{\left(S_{A_{observed}} - S_{AmCDW}\right) \times \left(\Theta_{WW} - \Theta_{mCDW}\right)}{\left(S_{AWW} - S_{AmCDW}\right)}}{\Theta_{meltwater} - \Theta_{mCDW} - \frac{\left(S_{Ameltwater} - S_{AmCDW}\right) \times \left(\Theta_{WW} - \Theta_{mCDW}\right)}{\left(S_{AWW} - S_{AmCDW}\right)}} \quad (1)$$

where $\varphi_{meltwater}$ is the meltwater fraction (in g kg$^{-1}$), and $\Theta$ and $S_A$ with subscripts are endpoints of the water masses (*mCDW* modified Circumpolar Deep Water, *WW* Winter Water). The glacial meltwater endpoints ($\Theta = -90.8\,°C$ and $S_A = 0\,g\,kg^{-1}$) are consistent with Zheng et al.[18] and Biddle et al.[20] and do not vary over time. mCDW is the warmest water mass existing below about 400 m in our study region (Supplementary Fig. 2). We chose the mCDW endpoints ($\Theta = 1.055\,°C$ and $S_A = 34.864\,g\,kg^{-1}$) from the warmest water mass measured by the deepest sensor attached on the AMIGOS system. During the study period, AMIGOS measurements showed that the mCDW properties did not change considerably (Fig. 1c, d). Winter Water is formed during winter, when air-sea interactions cause surface cooling, wind stirring and brine rejection in the deepened mixed layers. The resultant Winter Water has a temperature near the freezing point and a salinity that is higher than summertime mixed layer. We therefore chose the Winter Water endpoints to be $\Theta = -1.860\,°C$ and $S_A = 34.250\,g\,kg^{-1}$ based on the seal-tag observations from winter 2019. We used the same endpoints to calculate meltwater content from the seal-tag data in 2020 within PIB.

The uncertainty induced by the accuracy of the AMIGOS hydrographic measurements was estimated using a 5000-cycle Monte Carlo simulation. For each cycle, meltwater content was recalculated after two hundred hydrographic measurements were perturbed with randomly-generated, normally distributed noise varying up to the largest MicroCAT measurement uncertainties ($\pm0.002\,°C$ for temperature and $\pm0.003\,g\,kg^{-1}$ for salinity, following the manufacturer's manual). The estimated uncertainty is $\pm0.2\,g\,kg^{-1}$ for the calculated meltwater content caused by measurement uncertainties. We further estimate the uncertainty induced by the Winter Water endpoint chosen. The Winter Water endpoints were set to be the surface freezing point, and the absolute salinity was set to be a value between $34.20\,g\,kg^{-1}$ and $34.32\,g\,kg^{-1}$. While the lower bound falls closer to the Winter Water core measured by the seal tags, the upper bound is the Winter Water value found in PIB[18]. Varying the Winter Water endpoints changes slightly the calculated meltwater contents, but the temporal pattern of meltwater content remains the same (Supplementary Fig. 6). These uncertainties are negligible in our study region and are unlikely to cause qualitative change to our results.

## Temperature flux calculation

Along-channel conservative temperature flux ($\Theta_{flux}$; MW m$^{-2}$) for each set of MicroCAT and Aquadopp sensors is calculated using the following equation:

$$\Theta_{flux} = \rho \times c_p \times U \times (\Theta - \Theta_{fp}) \quad (2)$$

where $\rho$ is the seawater density (kg m$^{-3}$), $c_p$ is the specific heat capacity ($3991.9\,J\,kg^{-1}\,K^{-1}$), $U$ is the along-channel velocity (m s$^{-1}$) rotated 119° clockwise from east, $\Theta$ (K) is conservative temperature, and $\Theta_{fp}$ (K) is the conservative temperature at which seawater freezes at the surface. Positive $\Theta_{flux}$ is southwestward, i.e., into the cavity, whereas negative $\Theta_{flux}$ is northeastward, i.e., out of the cavity.

## Regional model simulation and particle release experiment

We use two regional configurations of the Massachusetts Institute of Technology general circulation model (MITgcm) from Nakayama et al.[11] and Nakayama et al.[44]. In Nakayama et al.[11], the model simulates circulation in the eastern Amundsen Sea with a nominal 200-m horizontal and 10-m vertical resolutions. Its bathymetry is based on the International Bathymetric Chart of the Southern Ocean (IBCSO)[45], with more accurate bathymetry for the region near Pine Island, and the Crosson and Dotson ice shelves[13,46]. The ice-shelf drafts were obtained using Antarctic Bedrock Mapping (BEDMAP-2; ref. 47) for Thwaites, Dotson, and Crosson ice shelves and high-resolution observations from commercial, sub-metre satellite stereo imagery for Pine Island Ice Shelf[48]. The ice-shelf thickness and cavity geometry were assumed to be in a steady-state. The model is forced at the lateral boundaries with a larger-domain model[44] and atmospheric forcing is ERA-Interim[49]. The model is run for 60 days with daily averaged outputs. See Nakayama et al.[11] for more information on the model.

In Nakayama et al.[44], the model simulates ocean circulation in the Amundsen and Bellingshausen Seas with a nominal 3–4 km horizontal resolution on monthly averaged outputs. We use extended model simulation based on Nakayama et al.[44] running from 1992 to 2020. Model bathymetry is based on the IBCSO[45], with more accurate bathymetry for the region near the Crosson and Dotson ice shelves[46]. The ice-shelf drafts were obtained using BEDMAP-2, ref. 47. We use this model output to investigate changes in simulated pycnocline for February–March, August–September, and December 2020.

Particle-release experiments are conducted offline using daily outputs of ocean current, from the Nakayama et al.[11] models, using Octopus (https://github.com/jinbow/Octopus). The particles are released beneath Pine Island Ice Shelf, in the area bounded by 102.073°–100.212°W and 75.355°–75.041°S (Fig. 5). To track particles that mix with glacial meltwater, the depth range is set to 195-495 m and potential density classes lighter than 1027.60 kg m$^{-3}$. For illustration purposes, the particles shown in Fig. 5 and Supplementary Movie 1 are

restricted to meltwater concentrations between 10 and 25 g kg$^{-1}$ and depth range between 250 and 400 m, based on day 2, to match the TEIS draft and avoid surface waters.

## Sea-ice concentration data

Daily mean sea-ice concentration data were obtained from Advanced Microwave Scanning Radiometer–Earth Observing System (AMSR-E) and Advanced Microwave Scanning Radiometer 2 (AMSR2) on a 3.125 km grid between 1 June 2002 and 1 April 2021 (ref. 50). We used the high-resolution vector polygons of the Antarctic coastline from the SCAR Antarctic Digital Database[51] to mask land and ice shelves in the AMSR-E/AMSR2 sea-ice concentration data. MODIS Terra Surface Reflectance 5-minute L2 Swath (MOD09 Band 4; ref. 52) from early and late summer in 2019–2020 and 2020–2021 were used to illustrate the sea-ice cover in PIB between the two summers.

## Data availability

All datasets used in this manuscript are stores in public repositories. The AMIGOS and borehole CTD datasets are available at the U.S. Antarctic Program Data Center repository (https://www.usap-dc.org/view/project/p0010162 and https://www.usap-dc.org/view/dataset/601623). The fibre-optic profiles are available at https://zenodo.org/record/7385608#.Y4h7UOzP1qs. The CTD measurements from 2020 and the seal-tag data are available at https://www.bodc.ac.uk/. The higher-resolution MITgcm code and daily outputs are available at https://ecco.jpl.nasa.gov/drive/files/ECCO2/High_res_PIG/AMS_200m/latlon_run8_tracer3_init2_cont_2. The relatively lower-resolution MITgcm code and monthly outputs are available at https://ecco.jpl.nasa.gov/drive/files/ECCO2/LLC1080_REG_AMS/Hyogo_et_al_2022. The AMSR-E/AMSR2 sea-ice concentration is available at https://seaice.uni-bremen.de/data/amsre/asi_daygrid_swath/s3125/ and https://seaice.uni-bremen.de/data/amsr2/asi_daygrid_swath/s3125/ under "Antarctic3125NoLandMask" folders.

## Code availability

The Matlab scripts used for the analysis described in this study can be obtained from the corresponding author upon request.

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

## Acknowledgements

We thank the scientists, technicians, and ship crew involved in the installation of the AMIGOS moorings and ocean data collection. This work is from the TARSAN project, a component of the International Thwaites Glacier Collaboration (ITGC; https://thwaitesglacier.org/). T.A.S., T.S., C.W., M.T., A.M., K.E.A., S.W.T., and E.P. thank the support from National Science Foundation (NSF) Grant 1929991. T.S.D., K.J.H., R.A.H., and A.K.W. thank the support from Natural Environment Research Council (NERC) Grant NE/S006419/1. L.B. and G.A.B. thank support from NERC Grant NE/S006591/1. Y.Z. acknowledges support from the China Scholarship Council, the University of East Anglia, and the European Research Council (under H2020-EU.1.1.; Grant 741120). Y.N. received support from the Grants-in-Aid for Scientific Research (19K23447, 21K13989) of the Japanese Ministry of Education, Culture, Sports, Science, and Technology. S.H. was supported by JST, the establishment of university fellowships towards the creation of science and technology innovation (Grant JPMJFS2101). Logistics were provided by NSF-U.S. Antarctic Program and NERC-British Antarctic Survey. ITGC Contribution No. ITGC-062.

## Author contributions

The study was conceptualized by T.S.D., K.J.H., and R.A.H. The AMIGOS fieldwork was done by T.A.S., C.W., M.T., A.M., E.P., K.E.A., and S.W.T. Seal-tagging and data quality control were done by L.B. and G.A.B. Methodology, data processing and analysis were done by T.S.D., Y.Z., and T.S. MITgcm simulations were performed by Y.N. and S.H. D.T.S. calibration was done by S.W.T. E.P., K.J.H., R.A.H., L.B., A.M., T.A.S., M.T., and A.K.W. were responsible for funding acquisition. T.S.D. wrote the original draft with input from all co-authors. T.S.D., K.J.H., R.A.H., T.A.S., Y.Z., Y.N., S.H., T.S., A.K.W., C.W., M.T., A.M., K.E.A., L.B., G.A.B., S.W.T., and E.P. revised and approved the final version of the manuscript.

## Competing interests

The authors declare that they have no conflict of interest.
