## [Peer Review File · Nature Communications]

Ocean variability beneath Thwaites Eastern Ice Shelf driven by the Pine Island Bay Gyre strengthREVIEWER COMMENTS

Reviewer #1 (Remarks to the Author):

Summary

The Paper of Dotto et al. uses two sub-ice shelf moorings, seal tag data and numerical simulations from MITgcm to explore the variability in ocean conditions beneath Thwaites Eastern Ice Shelf (TEIS) between January 2020 and April 2021. They link the variability in ocean conditions to variability in the strength of the gyre in Pine Island Bay, arguing that melt water sourced beneath Pine Island Ice Shelf feeds into the Thwaites Eastern Ice Shelf Cavity, while weakening of the Pine Island Bay gyre modified the isopycnal structure and brought warm waters closer to the ice base.

This is a unique data set that represents the first long-term observations from the main cavity beneath Thwaites Eastern Ice Shelf and is worthy of publication in Nature Communications. There are a number of areas, however, where the paper could be improved before it is ready for publication. The authors provide a convincing argument that the meltwater fraction beneath TESI increased throughout the observational period, and the meltwater can be sourced from beneath Pine Island Ice Shelf. The evidence that the mCDW layer warmed is less convincing, while the evidence of the mCDW layer thickening needs more emphasis in the main figures rather than the supplementals. I detail my major concerns below.

Major Comments

1. The authors argue in the abstract and throughout the paper (e.g. lines 106-107, 200-201) that the mCDW layer beneath TEIS warmed throughout the observational period. The evidence for this change, however, appears limited. As the authors state on lines 90-95, the core of mCDW is found below 650 m, and the temperature of this layer was relatively constant throughout the observational record (Figure 1c). Close examination of Supplementary Figure 4 shows a very modest warming of 0.1 deg C in T/S space at the lower sensors, although when plotted against the CTD casts (Supplementary Figure 2), the time average is visually no higher than the CTD temperature. Therefore it is hard to argue that a significant change in the properties of the mCDW layer has occurred. There is a large increase in temperature at the upper sensor at AMIGOS3c, however this sensor is not sampling the mCDW layer (it is positioned above the top of the mCDW thermocline) and instead this temperature increase is associated with the change in melt water fraction rather than a change in the mCDW properties (as the authors argue on lines 109-110). The authors should be careful to make this distinction in the main text. Currently lines 106-109 give the impression that the change in temperature in the upper layer at AMIGOS3c is due to the change in mCDW temperature, which I don't believe is the correct interpretation.

2. The authors claim that the mCDW layer has thickened through the observational period, yet evidence for this change is lacking in the main figures. A thicker mCDW layer should raise the height of the thermocline, driving strong variability in temperature and salinity between 400 and 650 m (the depth range of the thermocline). The upper sensor at AMIGOS3a is situated in the middle of the mCDW thermocline and should therefore show a trend towards warmer and saltier conditions. Careful examination of Supplementary Figure 4e shows that this is the case, but the trend is entirely masked in Figure 1c and 1d due to the wide range of the axes. I would like to see clearer evidence of this trend in the main figures to support the authors assertion that the mCDW layer has thickened. Can the authors also comment on why the temporal variability observed throughout the observational period for the upper sensor at AMIGOS3a is not significantly larger than the variability seen between the two CTD casts (Supplementary Figure 2)?

3. The authors present a time series of vertical displacement from the upper sensor at

AMIGOS3c in Supplementary Figure 3 and argue that vertical displacement can explain part of the observed temporal variability (lines 99-102). These vertical displacements, however, only represent the isopycnal depth change that is required to explain the perturbations from the time mean density given the background density profile. Instead, the authors should combine these vertical displacements with the background temperature/salinity gradient to calculate how much T/S can vary due to isopycnal heave, and then compare this with the observed variability in T/S to determine whether vertical displacement can explain the signals in temperature and salinity. This analysis will justify the assertion that authors make on lines 181-183 that the hydrographic variability can be explained by vertical displacement of the isopycnals.

4. Figure 1e provides convincing evidence that the meltwater fraction in the upper layers beneath TEIS increased throughout the observational period. The authors argue that the lack of basal melting observed at the AMIGOS sites suggests that the melt water was advected into the cavity from Pine Island Bay (lines 111-117). Can the authors fully rule out the possibility that the melt water has been sourced from deeper in the TESI cavity, and has been advected up through the basal channel towards the AMIGOS sites?

5. Flow at the upper sensors on both AMIGOS moorings is directed towards the southwest, and the authors argue that this suggests an inflow of waters from PIB (lines 118-123). It's not clear to me that the basal channel is playing a large role in facilitating this inflow, however. If the inflow was truly barotropic and required the basal channel to provide contours of constant water column thickness, the southwestward flow at AMIGOS3c should be significantly larger than that at AMIGOS3a. Instead, the time mean flow speed at AMIGOS3a is larger than that at AMIGOS3c. This suggests that the inflow is predominantly baroclinic, and the ice front presents no topographic barrier to the inflow (lines 123-129). Can the authors use the output of the MITgcm simulations to determine the fraction of the inflow that is baroclinic vs. barotropic?

6. On lines 176-181 the authors describe the large scale changes in temperature, salinity, melt water content and density in the upper layer. Can the authors clarify whether they are referring just to the data from the upper sensor at AMIGOS3c (which is in the upper layer), or whether they are also referring to data from the upper sensor at AMIGOS3a? If the latter is true, they need to make it clear that the upper sensor at AMIGOS3a is not in the upper layer – it is in the main mCDW thermocline.

As mentioned in point 2, the range of the axes in Figure 1 make it very difficult to see the changes that the authors are referring too. Figure 1 must be modified to ensure the changes in T/S at the different sensor depths can be easily visualised by the reader.

7. The authors provide evidence from numerical simulations that meltwater from PIIS can reach TEIS. This is a convincing argument; however, I am not clear how this analysis relates to their final section: Driving forces for the flow under the TEIS. On lines 193-195 they argue that a weaker gyre shoaled the isopycnals upwards which is consistent with the warmer conditions observed near the ice base throughout summer 2020-2021. Earlier in the paper, however, the authors argue that these warmer conditions are driven by meltwater enriched waters being advected into the cavity from PIB (lines 109-110). There appears to be some confusion between whether the changes they observe are advective or dynamic (i.e. isopycnal heave) in nature, and which processes are important at which depths in the water column. More work is needed here to clarify the authors thinking.

Lines 200-204 also cause similar confusion. There was a net warming of the water column in the observations, but I don't think it is correct to ascribe this warming solely to a change in isopycnal depth and the mCDW fraction. The authors need to be careful to clarify that the strongest warming signal at the upper sensor of AMIGOS3c is a meltwater driven signal, whereas the much smaller change in temperature at the upper sensor of AMIGOS3a is an mCDW signal driven by changes in the mCDW layer thickness. These confusions must be clarified before the paper is ready for publication.

Minor Comments

1. Line 38: "grounding line retreating rate" -> "grounding line retreat rate"
2. Line 39: "the global sea level" -> "global sea level"
3. Line 48: "the TEIS" -> "TEIS". This needs to be changed throughout the paper. There should be no "the" in front of Thwaites Eastern Ice Shelf. There should also be no "the" in front of Pine Island Bay or Pine Island Ice Shelf. There should be a "the" in front of Thwaites Eastern Ice Shelf cavity and PIB gyre.
4. Line 48: "ocean-circulation" -> "ocean circulation"
5. Line 62: "exported then" -> "then exported"
6. Line 63: "Although the PIIS meltwater-enriched waters are spread" -> "Although meltwater-enriched waters from PIIS are spread"
7. Line 64: "Idealised studies suggested" -> "Idealised studies suggest"
8. Line 65: "presence of lighter, freshened" -> "presence of lighter and fresher"
9. Line 65: "in the ice cavities" -> "in ice shelf cavities"
10. Line 66: "beneath the ice shelf" -> "beneath the ice base"
11. Line 66: "However, the effects from the interaction" -> "However, the effect"
12. Line 67: "the adjacent ice shelves" -> "adjacent ice shelves"
13. Line 72: "observed in the TEIS cavity" -> "observed beneath TEIS"
14. Line 73: "weakening or strengthening" -> "weakening and strengthening"
15. Line 74: "landfast-ice cover" -> "landfast sea ice cover"
16. Line 76: "control the ice-shelf-cavity temperatures" -> "control ice shelf cavity temperature"
17. Line 84: delete "small deformation and"
18. Line 86: delete "carved into the TEIS base"
19. Line 98: "beneath the TEIS" -> "beneath the ice base"
20. Line 102: instead of saying "winter 2020" please give months. This avoids any unnecessary confusion. Please also change on lines 178-180 and throughout the paper where required.
21. Line 118: "southwestward in" -> "southwestward at"
22. Line 118: "The speeds" -> "The speed"
23. Line 119: "were higher than" -> "was higher than"
24. Line 120: "and they were" -> "and the flow was"

25. Line 129: "in all depths" -> "at all depths"
26. Line 130: "buoyant plume at the ice-shelf ocean boundary layer" -> "buoyant plume within the ice-shelf ocean boundary layer"
27. Line 152: "the region where most of the PIIS melt water is exported" -> "the region where most meltwater is exported from beneath PIIS"
28. Line 154: "carved beneath the PIIS" -> "carved into PIIS"
29. Line 154: "flow westward with a coastal current" -> "flow westward in a coastal current"
30. Line 205: "originates at the PIIS" -> "originates beneath PIIS"
31. Line 211: "The rise of the isopycnals" -> "Raised isopycnals"
32. Line 213: "Summer 2020-2021" -> "Summer of 2020-2021"
33. Line 218: "interactions at" -> "interactions in"
34. Figure 5: there appears to be no schematic of the PIB gyre in the upper left panel. There is no yellow shading in panel c
35. Supplementary Figure 2a: is the black dashed line the in-situ freezing temperature?
36. Supplementary Figure 4: there are no thin cyan dashed line showing the thermohaline limits

Reviewer #2 (Remarks to the Author):

This paper presents new mooring data and finite element model simulation showing new aspects of ocean variability over the Thwaites eastern ice-shelf.

In particular, between January 2020 to March 2021 authors observe a thickening warm deep water with characteristics similar to modified CDW and suggest that meltwater from Pine Island Ice Shelf reaches the TEIS cavity and contributes to its net meltwater content, and the horizontal heat transport. The authors propose that the TEIS oceanographic variability in the cavity is linked to weakening or strengthening of the Pine Island Bay cyclonic gyre, caused by local variability in landfast-ice cover. Moreover, water masses infiltrate the central cavity below TEIS from the east, indicating that the east-west connection for the deep water.

This paper is well written (except for some quirks listed below), fits the topic of the journal and adds a good piece of novel information on the extended literature over TEIS and sheds further lights on ocean variability processes that could also characterize several other areas in Antarctica.

Several times authors refer to colors which appears to me (and to the rgb converter on my computer) not reported in the actual figure. As an example Figure 1 green and cyan arrows are very hard to distinguish and the explanation of what these arrows are is missing, or yellow shading highlights in figure 5, or "this cyan dashed line shows the thermohaline" which looks more like purple to me ?

I would ask the authors to add numbers or letters to lines wherever possible (or correct the wording if the wrong color was used) to help guiding color blind people.

L107-108 I would like to see an error bar on the temperature and salinity estimates

L111-113 please provide sizes (and errors)for layers changes

L114 please clarify if “data not shown” is meant to be there or additionally please add an explanation in the supplementary material

L145-146 please clarify how you measured the lack of melting events

Figure 1a black dashed arrows explanation missing

Figure 1b cyan arrow explanation missing are these max speed from figure 2?

Figure 4 modis image is barely visible

Figure 5 please use uniform style/position for the date-box

Figure 5b neural density definition not described anywhere in the manuscript

Figure 5c graphic chopped (top left)

Figure 5 “Yellow shading highlights”barely visible

Figure S1 please add details/ reference here of the ground penetrating radar data used in this study.

Figure s2 naming convention of the sub panels has a different structure and geometric order compared to the figures in the main text. I would uniform them for consistency. Same for figure s4.

Figure S4 cyan you mean purple?

Reviewer #3 (Remarks to the Author):

This paper investigates the ocean circulation below the Thwaites Eastern Ice Shelf (TEIS) using ice-shelf moorings from January 2020 to March 2021. Thwaites Glacier is among the fastest flowing marine-terminating glaciers in Antarctica and is the largest contributor to sea-level rise in Antarctica in the past decades; thus, its study is of high importance in the context of climate change. The authors find that the TEIS cavity has warmed during the period analyzed, and that the glacial meltwater content has increased. With the help of a high-resolution regional ocean model, they find that the Pine Island Ice Shelf constitutes a substantial source of meltwater found in the TEIS cavity. Finally, the authors suggest that the weakening of the Pine Island Bay gyre, caused by a relatively high sea-ice concentration, has brought warm waters to TEIS cavity.

The paper is very well written and very well structured. The authors convince the reader with strong evidence and nice figures (and movie) that the main processes identified are key for the identified changes. I recommend the paper for publication if the authors deal with three main comments I have:

1) It seems a bit strange to me that the increase in meltwater content in the TEIS cavity, as identified by the authors, has not led to an increase in basal melt (as stated in L114 and L145). I think the authors should discuss this interesting aspect in their paper.

2) I think the authors should discuss (in their Discussion) the potential effect of the process identified (increase in meltwater content coming from Pine Island Ice Shelf) on the fate of Thwaites Glacier: does it make the glacier more likely to melt faster in the future? What is the relative contribution of this process (compared to direct basal melt) to the overall melt of Thwaites Glacier?

3) The section on the driving forces for the flow under TEIS (L175-197) is not enough

backed up by evidence, especially because this is one the main results of the study (which appears in the title). Or, at least, there is not enough explanation about what is the exact process that led from the weakening of the gyre to warm water going to the TEIS cavity. Wouldn't it be possible to demonstrate that with the regional ocean model? As it appears in the text, this is a strong hypothesis and there is not enough evidence for this process. I suggest the authors to either provide stronger evidence (including a small modeling experiment) or to improve the explanation of the process if they think sufficient evidence is present.

I have two additional small comments:

L374: The hyperlink to the model does not seem to work.

Supplementary Figure 1b: It would be clearer to use longitudes from -180°W to +180°E instead of 0-360° to ease the comparison with Supplementary Figure 1a. Also, I am wondering if a color bar for the conservative temperature is really needed as only 4 points are shown for which the exact numbers are provided.

Response to the Reviewers

We are grateful to the Reviewers for their very helpful and constructive feedback. In the following, we outline how we have responded to their comments. Comments by the Reviewers are shown in *black*, and our responses in *blue*. Track-changes done in the revised manuscript are shown in *blue*.

Response to Reviewer #1:

This is a unique data set that represents the first long-term observations from the main cavity beneath Thwaites Eastern Ice Shelf and is worthy of publication in Nature Communications.

We thank the Reviewer for the time spent reading the paper and for the constructive comments provided.

There are a number of areas, however, where the paper could be improved before it is ready for publication. The authors provide a convincing argument that the meltwater fraction beneath TESI increased throughout the observational period, and the meltwater can be sourced from beneath Pine Island Ice Shelf. The evidence that the mCDW layer warmed is less convincing, while the evidence of the mCDW layer thickening needs more emphasis in the main figures rather than the supplementals. I detail my major concerns below.

We have used these suggestions for strengthening the paper in the new version of the manuscript and we hope that the answers below satisfy the concerns raised by the reviewer.

Major Comments

1. The authors argue in the abstract and throughout the paper (e.g. lines 106-107, 200-201) that the mCDW layer beneath TEIS warmed throughout the observational period. The evidence for this change, however, appears limited. As the authors state on lines 90-95, the core of mCDW is found below 650 m, and the temperature of this layer was relatively constant throughout the observational record (Figure 1c). Close examination of Supplementary Figure 4 shows a very modest warming of 0.1 deg C in T/S space at the lower sensors, although when plotted against the CTD casts (Supplementary Figure 2), the time average is visually no higher than the CTD temperature. Therefore it is hard to argue that a significant change in the properties of the mCDW layer has occurred. There is a large increase in temperature at the upper sensor at AMIGOS3c, however this sensor is not sampling the mCDW layer (it is positioned above the top of the mCDW thermocline) and instead this temperature increase is associated with the change in melt water fraction rather than a change in the mCDW properties (as the authors argue on lines 109-110). The authors should be careful to make this distinction in the main text. Currently lines 106-109 give the impression that the change in temperature in the upper

layer at AMIGOS3c is due to the change in mCDW temperature, which I don't believe is the correct interpretation.

Thank you for spotting this ambiguity, we agree with the reviewer and have revised the text. Instead of saying that the mCDW warmed during the period, we now explain that the recorded temperature increase in the upper sensors of AMIGOS3a (~316 m) and AMIGOS3c (~521m) is due to the increased meltwater concentration in the thermocline layer.

2. The authors claim that the mCDW layer has thickened through the observational period, yet evidence for this change is lacking in the main figures. A thicker mCDW layer should raise the height of the thermocline, driving strong variability in temperature and salinity between 400 and 650 m (the depth range of the thermocline). The upper sensor at AMIGOS3a is situated in the middle of the mCDW thermocline and should therefore show a trend towards warmer and saltier conditions. Careful examination of Supplementary Figure 4e shows that this is the case, but the trend is entirely masked in Figure 1c and 1d due to the wide range of the axes. I would like to see clearer evidence of this trend in the main figures to support the authors assertion that the mCDW layer has thickened. Can the authors also comment on why the temporal variability observed throughout the observational period for the upper sensor at AMIGOS3a is not significantly larger than the variability seen between the two CTD casts (Supplementary Figure 2)?

Thank you for drawing our attention to this. We agree with the reviewer's interpretation that the observed changes over the measurement period are due to changes in the thermocline waters rather than the mCDW layer properties. We have revised the manuscript in several places to make this clearer (e.g., Lines 268-278). We also decided to move the TS-time diagrams to the manuscript as our new Fig. 2 to allow the reader to visualise the thermohaline changes better. In the new version of the manuscript, we included a schematic (new Fig. 8) to exemplify better the processes described here.

Regarding the question about the difference between the two CTD casts and the variability seen in the time series: it is not clear to us why the reviewer expects much larger variability than the difference between the two casts. The two casts were not taken at the same time nor the same location. The variability observed in the profiles could be related to vertical movements of the thermocline in the order of tens of metres (Supplementary Fig. 2), which agrees with the variability observed in Supplementary Fig. 3. A comment to explain this has been included in Lines 104-109.

3. The authors present a time series of vertical displacement from the upper sensor at AMIGOS3c in Supplementary Figure 3 and argue that vertical displacement can explain part of the observed temporal variability (lines 99-102). These vertical displacements, however, only represent the isopycnal depth change that is required to explain the perturbations from the time mean density given the background density profile. Instead, the authors should combine these vertical displacements with the background temperature/salinity gradient to calculate how much T/S can vary due to isopycnal heave, and then compare this with the observed variability in T/S to determine whether vertical displacement can explain the signals in temperature and salinity. This analysis

will justify the assertion that authors make on lines 181-183 that the hydrographic variability can be explained by vertical displacement of the isopycnals.

This is exactly what we have done. We are sorry that it was not clearer before. The vertical displacements shown in Supplementary Fig. 3a were calculated based on the gradients of density (isopycnal), temperature-only (isotherm), and salinity-only (isohaline) from the borehole CTD combined with the temporal variations of the water properties from the mooring (see methods). Therefore, the temporal variations observed in temperature and salinity seen in AMIGOS3c-Upper are already accounted for in the time series of isotherm and isohaline vertical displacements shown in Supplementary Fig. 3a. Using the “*vertical displacements with the background temperature/salinity gradient to calculate how much T/S can vary due to isopycnal heave*” would give us the same values from Fig. 1c,d, which is redundant. We clarify the concern raised by the reviewer in the revised version (Lines 106-109).

4. Figure 1e provides convincing evidence that the meltwater fraction in the upper layers beneath TEIS increased throughout the observational period. The authors argue that the lack of basal melting observed at the AMIGOS sites suggests that the melt water was advected into the cavity from Pine Island Bay (lines 111-117). Can the authors fully rule out the possibility that the melt water has been sourced from deeper in the TESI cavity, and has been advected up through the basal channel towards the AMIGOS sites?

The reviewer is correct that we cannot definitively rule out the possibility of the advection of meltwater from deeper in TEIS cavity. To do so would require full knowledge of the bathymetry (seabed and ice base) and the circulation in the cavity. However, the westward flow of Pine Island Bay water is the most likely explanation because the flow velocity underneath TEIS is along the basal channel southwestward (Figs. 1b and 4) not northeastward as would be required for flow from near the grounding line. Further supporting evidence comes from the hydrographic analysis of Wåhlin et al. (2021), showing that deep and intermediate water has moved from Pine Island Bay, below TEIS, and into the region near and underneath the northern opening. Together, these indicate that the thermocline water is indeed imported from Pine Island Bay. In order to clarify this point, the section has been rewritten (Lines 152-167). We have also added a new figure in supplementary material where temperature and salinity data from the 2019 cruise are plotted together with the mooring data to illustrate that the water masses found in Pine Island Bay are the same as those found underneath TEIS at the mooring sites (new Supplementary Fig. 4).

5. Flow at the upper sensors on both AMIGOS moorings is directed towards the southwest, and the authors argue that this suggests an inflow of waters from PIB (lines 118-123). It's not clear to me that the basal channel is playing a large role in facilitating this inflow, however. If the inflow was truly barotropic and required the basal channel to provide contours of constant water column thickness, the southwestward flow at AMIGOS3c should be significantly larger than that at AMIGOS3a. Instead, the time mean flow speed at AMIGOS3a is larger than that at AMIGOS3c. This suggests that the inflow is predominantly baroclinic, and the ice front presents no topographic barrier to the inflow (lines 123-129). Can the authors use the output of the MITgcm simulations to determine the fraction of the inflow that is baroclinic vs. barotropic?

We agree with the reviewer regarding the flow being mainly baroclinic, and we now discuss the arguments provided by the reviewer in Lines 145-151. Although knowing whether the flow below the ice shelf is mainly baroclinic or barotropic is a good question, we think it is not pertinent for the general discussion and findings of the paper. Therefore, we decided to remove from the manuscript any mention regarding the flow being steered by the basal channel. However, we note that it is quite interesting that the flow has a similar direction to the sub-ice shelf channel (e.g., Fig. 3a-d).

6. On lines 176-181 the authors describe the large scale changes in temperature, salinity, melt water content and density in the upper layer. Can the authors clarify whether they are referring just to the data from the upper sensor at AMIGOS3c (which is in the upper layer), or whether they are also referring to data from the upper sensor at AMIGOS3a? If the latter is true, they need to make it clear that the upper sensor at AMIGOS3a is not in the upper layer – it is in the main mCDW thermocline. As mentioned in point 2, the range of the axes in Figure 1 make it very difficult to see the changes that the authors are referring to. Figure 1 must be modified to ensure the changes in T/S at the different sensor depths can be easily visualised by the reader.

We reword the sentence to “*The seawater properties at the upper sensor of AMIGOS3c, which is in the upper layer, were distinct in different seasons (Fig 1c-e).*” to make it clear that we were referring to the upper layer at the AMIGOS3c. The axes of Fig. 1 were modified to better visualise the temporal changes in the deep waters (see the revised manuscript).

7. The authors provide evidence from numerical simulations that meltwater from PIIS can reach TEIS. This is a convincing argument; however, I am not clear how this analysis relates to their final section: Driving forces for the flow under the TEIS. On lines 193-195 they argue that a weaker gyre shoaled the isopycnals upwards which is consistent with the warmer conditions observed near the ice base throughout summer 2020-2021. Earlier in the paper, however, the authors argue that these warmer conditions are driven by meltwater enriched waters being advected into the cavity from PIB (lines 109-110). There appears to be some confusion between whether the changes they observe are advective or dynamic (i.e. isopycnal heave) in nature, and which processes are important at which depths in the water column. More work is needed here to clarify the authors thinking.

Lines 200-204 also cause similar confusion. There was a net warming of the water column in the observations, but I don't think it is correct to ascribe this warming solely to a change in isopycnal depth and the mCDW fraction. The authors need to be careful to clarify that the strongest warming signal at the upper sensor of AMIGOS3c is a meltwater driven signal, whereas the much smaller change in temperature at the upper sensor of AMIGOS3a is an mCDW signal driven by changes in the mCDW layer thickness. These confusions must be clarified before the paper is ready for publication. We agree with the reviewer that the explanations caused some confusion. To make these points better explained, we added a few sentences in the revised manuscript. Our datasets suggest that both processes (i.e., an increase of meltwater and shoaling of isopycnals beneath TEIS) are happening. The increase of meltwater beneath TEIS is

responsible for the longer-term warming shown in Fig. 1c (as described in Sections “Interannual variability in ocean conditions beneath TEIS” and “Meltwater pathways between PIIS and TEIS”). However, there are periods when the meltwater alone cannot totally explain the warming signal, such as between August and September 2020, when temperature, salinity and density rose (Fig. 1c-d and Supplementary Fig. 3b), but the meltwater content did not increase (Fig. 1e). During those months, we attributed the changes in the water masses due to the heave of the isopycnals caused by the expansion of the thermocline bringing deeper water masses upwards. Additionally, the long-lasting spin-down of the PIB gyre due to long periods of sea-ice-cover might have reduced the export of meltwater-enriched waters off the PIB region. More meltwater in the system and the flattening of the isopycnals contributed to this heat accumulated in the area to be transported into the TEIS cavity. We rewrote many parts of that section (Lines 219-264) to make these explanations clearer. This section has also a schematic (new Fig. 8) to visualise the processes discussed in different periods. We also make this distinction clearer in the “Discussion” by pointing out that most of the warming in the upper layers is meltwater-driven (Lines 268-270).

Minor Comments

1. Line 38: “grounding line retreating rate” -> “grounding line retreat rate”

Done

2. Line 39: “the global sea level” -> “global sea level”

Done

3. Line 48: “the TEIS” -> “TEIS”. This needs to be changed throughout the paper. There should be no “the” in front of Thwaites Eastern Ice Shelf. There should also be no “the” in front of Pine Island Bay or Pine Island Ice Shelf. There should be a “the” in front of Thwaites Eastern Ice Shelf cavity and PIB gyre.

Thank you. The changes were made.

4. Line 48: “ocean-circulation” -> “ocean circulation”

Done

5. Line 62: “exported then” -> “then exported”

Done

6. Line 63: “Although the PIIS meltwater-enriched waters are spread” -> “Although meltwater-enriched waters from PIIS are spread”

Done

7. Line 64: “Idealised studies suggested” -> “Idealised studies suggest”

Done

8. Line 65: “presence of lighter, freshened” -> “presence of lighter and fresher”

Done

9. Line 65: “in the ice cavities” -> “in ice shelf cavities”

Done

10. Line 66: “beneath the ice shelf” -> “beneath the ice base”

Done

11. Line 66: “However, the effects from the interaction” -> “However, the effect”

Done

12. Line 67: “the adjacent ice shelves” -> “adjacent ice shelves”

Done

13. Line 72: “observed in the TEIS cavity” -> “observed beneath TEIS”

Done

14. Line 73: “weakening or strengthening” -> “weakening and strengthening”

Done

15. Line 74: “landfast-ice cover” -> “landfast sea ice cover”

Done

16. Line 76: “control the ice-shelf-cavity temperatures” -> “control ice shelf cavity temperature”

Done

17. Line 84: delete “small deformation and”

Done

18. Line 86: delete “carved into the TEIS base”

Done

19. Line 98: “beneath the TEIS” -> “beneath the ice base”

Done

20. Line 102: instead of saying “winter 2020” please give months. This avoids any unnecessary confusion. Please also change on lines 178-180 and throughout the paper where required.

Done

21. Line 118: “southwestward in” -> “southwestward at”

Done

22. Line 118: “The speeds” -> “The speed”

Done

23. Line 119: “were higher than” -> “was higher than”

Done

24. Line 120: “and they were” -> “and the flow was”

Done

25. Line 129: “in all depths” -> “at all depths”

Done

26. Line 130: “buoyant plume at the ice-shelf ocean boundary layer” -> “buoyant plume within the ice-shelf ocean boundary layer”

Done

27. Line 152: “the region where most of the PIIS melt water is exported” -> “the region where most meltwater is exported from beneath PIIS”

Done

28. Line 154: “carved beneath the PIIS” -> “carved into PIIS”

Done

29. Line 154: “flow westward with a coastal current” -> “flow westward in a coastal current”

Done

30. Line 205: “originates at the PIIS” -> “originates beneath PIIS”

Done

31. Line 211: “The rise of the isopycnals” -> “Raised isopycnals”

Done

32. Line 213: “Summer 2020-2021” -> “Summer of 2020-2021”

Done

33. Line 218: “interactions at” -> “interactions in”

Done

34. Figure 5: there appears to be no schematic of the PIB gyre in the upper left panel. There is no yellow shading in panel c

We apologise for this issue. We just checked the PDF created during the submission and the schematic of the PIB gyre and the yellow shadings are not shown indeed. They are however in the word document. We will double-check this during the resubmission. The final figures in PDF format to be sent have the schematic of the PIB gyre and the yellow shadings.

35. Supplementary Figure 2a: is the black dashed line the in-situ freezing temperature? Yes. We added this information to the caption.

36. Supplementary Figure 4: there are no thin cyan dashed line showing the thermohaline limits

Thank you for indicating this. We removed any mention of the cyan rectangles in the new version.

Response to Reviewer #2:

This paper is well written (except for some quirks listed below), fits the topic of the journal and adds a good piece of novel information on the extended literature over TEIS and sheds further lights on ocean variability processes that could also characterize several other areas in Antarctica.

We thank the Reviewer for their constructive comments.

Several times authors refer to colors which appears to me (and to the rgb converter on my computer) not reported in the actual figure. As an example Figure 1 green and cyan arrows are very hard to distinguish and the explanation of what these arrows are is missing, or yellow shading highlights in figure 5, or “this cyan dashed line shows the thermohaline” which looks more like purple to me ? I would ask the authors to add numbers or letters to lines wherever possible (or correct the wording if the wrong color was used) to help guiding color blind people.

Thank you for the advice. We revisit the figures' colours to make them colourblind-friendly. Please, let us know if any other figure needs correction.

L107-108 I would like to see an error bar on the temperature and salinity estimates. These numbers show the order of magnitude of the changes in temperature and salinity between the beginning and the end of the record. They are not trends. Therefore, adding an error bar is irrelevant in this case. In the revised manuscript, we included the standard deviations to the time mean estimates in Lines 98-103.

L111-113 please provide sizes (and errors)for layers changes

We rewrote the sentence and now provide the magnitude of the meltwater content in the layers (see Lines 125-126). Note that the error associated with the meltwater estimate ($\pm 0.2 \text{ g kg}^{-1}$) is stated in the methods.

L114 please clarify if “data not shown” is meant to be there or additionally please add an explanation in the supplementary material

We removed “data not shown” and added a new Fig 3 that shows temperature profiles (monthly averaged) from Distributed Temperature Sensing (DTS). The DTS is a fibre-optic cable installed along the AMIGOS moorings. The DTS estimate temperature within the ice shelf and at the ocean, and the ice-ocean interface can be estimated by looking at the changes in the temperature gradient. Our new Fig. 3 inserted in the manuscript shows that the ice-ocean interface remained virtually steady when compared April 2020,

September 2020, and February 2021. Note that in April 2020, the borehole was still refreezing. See Lines 126-132.

L145-146 please clarify how you measured the lack of melting events

We measured the lack of large melting through identifying the ice-ocean interface from fibre-optic cables installed along the AMIGOS (Lines 126-132). Lines 356-368 (Methods) explain the fibre-optic cables.

Figure 1a black dashed arrows explanation missing

The explanation is now provided in the revised caption.

Figure 1b cyan arrow explanation missing are these max speed from figure 2?

The arrows in Fig. 1b are the time-mean velocity vectors. The explanation was already stated in the Figure caption "*Time-mean (January 2020 to March 2021) velocity vectors are shown for the upper sensors (cyan arrows) and deeper sensors (orange arrows) at the AMIGOS3a (blue dot) and AMIGOS3c (red dot) sites*". In the new figure, we changed the colour green of the arrows to orange.

Figure 4 modis image is barely visible

Figure 6a (previous Fig. 4) shows a period covered by sea ice, and it is the same as in Fig. 7d. There is nothing we can do to make it more visible. The land and ice shelf contours were provided to give a reference for the reader.

Figure 5 please use uniform style/position for the date-box

Done

Figure 5b neutral density definition not described anywhere in the manuscript

The reference for neutral density definition and calculation was added in Line 349.

Figure 5c graphic chopped (top left)

We apologise for this issue. The PDF created in the submission process chopped the graphic. The figure is not chopped in the word document. We will double-check this during the resubmission.

Figure 5 "Yellow shading highlights" barely visible

We apologise for this issue. We just checked the PDF created during the submission and the yellow shadings are not shown. They are present in the word document. We will double-check this during the resubmission.

Figure S1 please add details/ reference here of the ground penetrating radar data used in this study.

We have added details within the figure caption (See Supplementary Fig. 1) "*White line is the ice shelf draft as observed by ground-penetrating radar in December 2019 (standard processing included stacking variable from 16-32, band pass filtering range approximately 30 to 150 MHz, gain based on mean attenuation of signal, no migration).*"

Figure s2 naming convention of the sub panels has a different structure and geometric order compared to the figures in the main text. I would uniform them for consistency. Same for figure s4.

Thank you for the suggestion. We corrected the figures to keep consistency.

Figure S4 cyan you mean purple?

We removed any mention of the cyan rectangles.

Response to Reviewer #3:

The paper is very well written and very well structured. The authors convince the reader with strong evidence and nice figures (and movie) that the main processes identified are key for the identified changes. I recommend the paper for publication if the authors deal with three main comments I have:

We thank the Reviewer for their constructive comments. Below we address each question raised by the reviewer.

1) It seems a bit strange to me that the increase in meltwater content in the TEIS cavity, as identified by the authors, has not led to an increase in basal melt (as stated in L114 and L145). I think the authors should discuss this interesting aspect in their paper.

We agree with the reviewer and also find this very interesting. However, after thinking carefully about this, we are of the opinion that it is hard to use the comparison of the local mooring data with the integrated and comparatively low-resolution remote sensing data as proof. We introduce a possible physical process that can explain the observation, but have tried to be clear that we do not present evidence for this process. We speculate that the meltwater-enriched water from PIB might increase the stratification near the ice base as it replaces the relatively cold, salty and well-mixed Winter Water. The stronger stratification, possibly in combination with slow baroclinic ocean circulation, small tides, and low turbulence levels, inhibit the heat flux from the ocean to the ice base. However, since we have no sensors at the ice base this remains speculation. A paragraph has been added to the discussion in Lines 312-321.

2) I think the authors should discuss (in their Discussion) the potential effect of the process identified (increase in meltwater content coming from Pine Island Ice Shelf) on the fate of Thwaites Glacier: does it make the glacier more likely to melt faster in the future? What is the relative contribution of this process (compared to direct basal melt) to the overall melt of Thwaites Glacier?

This is a very good question, and we have added a short paragraph discussing this in the Discussion (Lines 307-314). Since we do not have data points right at the ice base, we cannot learn anything about the boundary layer process itself. Hence, we cannot say if this means that TEIS will melt faster in the future.

3) The section on the driving forces for the flow under TEIS (L175-197) is not enough backed up by evidence, especially because this is one the main results of the study

(which appears in the title). Or, at least, there is not enough explanation about what is the exact process that led from the weakening of the gyre to warm water going to the TEIS cavity. Wouldn't it be possible to demonstrate that with the regional ocean model? As it appears in the text, this is a strong hypothesis and there is not enough evidence for this process. I suggest the authors to either provide stronger evidence (including a small modeling experiment) or to improve the explanation of the process if they think sufficient evidence is present.

Thank you for the suggestion. We now use another ocean simulation, an updated model from Nakayama et al. 2018 (doi: 10.1038/s41467-018-05813-1), to demonstrate the temporal variations in the structure of the PIB gyre regarding sea-ice-free and sea-ice-covered periods. The new Figure 7 shows a section cross-cutting the PIB gyre (panel a) from which we plotted, in panel b, the isopycnals for Feb.-Mar. 2020 (blue), Aug.-Sep. 2020 (red) and Dec. 2020 (black dotted). This panel shows that the isopycnals change from highly tilted (and thus intensified PIB gyre) near TEIS in Feb.-Mar. 2020 (sea-ice-free) to flatten and weaker gyre in Aug.-Sep. 2020 (sea-ice-covered). The order of magnitude of the changes in the isopycnal depth (i.e., tens of metres; panel c) between Feb.-Mar. and Aug.-Sep. 2020 below TEIS agrees with the isopycnal displacement shown in Supplementary Fig. 3a. This variation in the density structure of the gyre agrees with observations (panel d). In Dec. 2020, the last month of the simulation and a period when the sea-ice did not melt completely (Fig. 9a), the isopycnals are flattened, but the density is reduced within the gyre. We attributed this reduction in density due to high meltwater that entered the system, and it was not exported from PIB due to a weakened PIB gyre. By deepening the isopycnals near and below TEIS, a higher volume of highly-meltwater-enriched waters from the upper ocean can access the cavity, which explains the increase in the meltwater content after Nov. 2020 (Fig. 1e). We discuss better this process in Lines 219-264.

I have two additional small comments:

L374: The hyperlink to the model does not seem to work.

We updated the hyperlink to point to the proper ocean simulation outputs (https://ecco.jpl.nasa.gov/drive/files/ECCO2/High_res_PIG/AMS_200m/latlon_run8_tracer3_init2_cont_2). The user needs to register at <https://urs.earthdata.nasa.gov/> prior downloading the data.

Supplementary Figure 1b: It would be clearer to use longitudes from -180°W to $+180^{\circ}\text{E}$ instead of $0-360^{\circ}$ to ease the comparison with Supplementary Figure 1a. Also, I am wondering if a color bar for the conservative temperature is really needed as only 4 points are shown for which the exact numbers are provided.

Done.

REVIEWERS' COMMENTS

Reviewer #1 (Remarks to the Author):

Summary

This is my second review of "Ocean variability beneath Thwaites Eastern Ice Shelf driven by the Pine Island Bay Gyre strength" by Dotto et al. The authors have responded to my comments satisfactorily, and barring some minor comments below, I can recommend this paper for publication.

Minor Comments

Line 101: "On occasion, water with mCDW characteristics was observed as shallow as the upper sensor on 102 AMIGO3a (~521 m)". I'm not 100% sure what evidence I should be looking at to see that this is the case. On the T/S diagram (Figure 2b,e), there doesn't appear to be any evidence the properties at AMIGOS3a become mCDW-like in nature. Also, "AMIGO" should be "AMIGOS" on line 102

Line 107-109: I remain unconvinced by the authors argument. They state in the text that they calculate the vertical displacements in each property that would be required to explain the observed variations in the AMIGOS3c-Upper time series. There is no evidence presented that these displacements are actually observed. I would much prefer to see a time series of actual displacements (in density space), and the size of the T/S variability that these observed displacements could drive.

Line 119: the upper layer of the TEIS cavity warmed – not the whole cavity.

Line 144: "westward flow" -> "southwestward flow"

Line 150: "AMIGOS3c would" -> "AMIGOS3c should"

Line 156: can you be more precise about what you mean by the "northern opening"

Line 287-190: "Our hypothesis implies, somewhat counter-intuitively, that these ice-shelf cavities will warm during cold ocean periods if that condition supports landfast sea-ice cover in PIB and adjacent to TEIS, by ultimately allowing more meltwater to enter the ice shelf cavities (Fig. 2c)". I'm not sure how true this is. Yes, the gyre will be modified to allow more melt water to enter ice shelf cavities, but there will also be less melt water (as PIIS will be melting more slowly) so the amount of heat available will be less.

Reviewer #2 (Remarks to the Author):

The Authors have addressed all my comments.
I recommend the editor to accept this manuscript.

Reviewer #3 (Remarks to the Author):

I congratulate the authors for responding to my comments with such a great care. All my concerns have been dealt with. I really appreciate the addition of the new Figures 7 and 8, which constitute important additions to this study. The paper has been much improved, especially the part about the driving forces and the discussion, and presents convincing evidence, so I recommend it for publication. There is a lot of potential for follow-up studies, e.g. 1) trying to better understand the link between meltwater content and basal melt under TEIS (my first main comment), and 2) the potential of the identified process in affecting the future melting of Thwaites Glacier.

Response to the Editor and Reviewers

We are grateful to the Editor and Reviewers for their very helpful and constructive feedback. In the following, we outline how we have responded. Comments by the Reviewers are shown in *black*, and our responses in *blue*. Track-changes done in the revised manuscript are shown in *blue*.

Response to Reviewer #1:

This is my second review of “Ocean variability beneath Thwaites Eastern Ice Shelf driven by the Pine Island Bay Gyre strength” by Dotto et al. The authors have responded to my comments satisfactorily, and barring some minor comments below, I can recommend this paper for publication.

We thank the Reviewer for the time spent reading the paper and for the constructive comments provided.

Minor Comments

Line 101: “On occasion, water with mCDW characteristics was observed as shallow as the upper sensor on 102 AMIGO3a (~521 m)”. I’m not 100% sure what evidence I should be looking at to see that this is the case. On the T/S diagram (Figure 2b,e), there doesn’t appear to be any evidence the properties at AMIGOS3a become mCDW-like in nature. Also, “AMIGO” should be “AMIGOS” on line 102

We agree with the reviewer that this is not “mCDW-like” in nature; we meant only that it was relatively warm and salty. Therefore, we rephrased this sentence to remove sentences mentioning mCDW for the AMIGOS3a-Upper. Instead, we re-wrote the sentence in Lines 100-102 to “A transitional layer between the warm and salty waters at deeper depths and cold and fresh waters at shallower depths was observed at the upper sensor on AMIGOS3a (~521 m; Supplementary Fig. 2) (...)”. With this, we avoid calling it by a water mass nomenclature, but keep the meaning of the sentence.

AMIGOS was corrected in Line 102.

Line 107-109: I remain unconvinced by the authors argument. They state in the text that they calculate the vertical displacements in each property that would be required to explain the observed variations in the AMIGOS3c-Upper time series. There is no evidence presented that these displacements are actually observed. I would much prefer to see a time series of actual displacements (in density space), and the size of the T/S variability that these observed displacements could drive.

Thank you for the suggestion. After some discussion amongst the authors and considering the confusion of the reviewer, we decided to make the calculation of vertical isopycnal displacement more robust and easier for our readers to understand. These new calculations more strongly and clearly support the (unchanged) conclusion that

vertical displacements are not able to account for the variations we see in temperature and salinity at the mooring.

Calculating absolute vertical displacement of water masses is not trivial without continuous vertical profiles of density/temperature/salinity. Additionally, to measure absolute vertical displacements from moorings would require many microcats closely spaced in the vertical. We have only the profile obtained on deployment, and the time series at the two single depths. Therefore, we calculate the vertical isopycnal displacement as follows. For each time step in the moored time series at AMIGOS3c-Upper, we determine the depth in the borehole profile that has a similar potential density to the mooring value. The vertical displacement is then the difference in depth between the mooring and the depth in the profile that has the same density. We determine this separately for potential density, absolute salinity and conservative temperature in order to have independent time series of vertical displacements for isopycnals, isohalines and isotherms. Because the temperature profile is not monotonic (Supplementary Figure 2a), early in the time series there are two solutions for the isotherm displacement above and below the mooring depth; we show these two solutions as dots. Later in the time series there is only one solution because the temperature is sufficiently warm that it is only found below the mooring. The resulting time series of vertical displacements are plotted in the new Supplementary Fig. 3a. That the three (independent) estimates of displacements do *not* agree supports our argument that the temporal variability seen at the mooring cannot be accounted for by vertical displacements alone.

Based on the reviewer's suggestion "*I would much prefer to see a time series of actual displacements (in density space), and the size of the T/S variability that these observed displacements could drive.*", we added an estimation of the values of temperature and salinity associated with the vertical displacements of density (black lines of the new Supplementary Fig. 3a). From this calculation, it is evident that variations in salinity are associated with the density incursions of water masses (Supplementary Fig. 3b), but the excess of heat observed over time cannot be explained by vertical movements of water masses (Supplementary Fig. 3c). Instead, it supports that advection of meltwater must be contributing to the increase of the heat content in the upper layers of the TEIS cavity.

We modified the explanation of the methods for calculating vertical displacements in Line 390-401. We also modified the description of these results in Lines 108-120.

Line 119: the upper layer of the TEIS cavity warmed – not the whole cavity.
Done.

Line 144: "westward flow" -> "southwestward flow"
Done.

Line 150: "AMIGOS3c would" -> "AMIGOS3c should"
Done.

Line 156: can you be more precise about what you mean by the "northern opening"

Done. We specify in the reviewed manuscript that the “northern opening” is the region “between TEIS and the Thwaites Ice Tongue” as described by Wåhlin et al. (2021) (Line 165).

Line 287-190: “Our hypothesis implies, somewhat counter-intuitively, that these ice-shelf cavities will warm during cold ocean periods if that condition supports landfast sea-ice cover in PIB and adjacent to TEIS, by ultimately allowing more meltwater to enter the ice shelf cavities (Fig. 2c)”. I’m not sure how true this is. Yes, the gyre will be modified to allow more melt water to enter ice shelf cavities, but there will also be less melt water (as PIIS will be melting more slowly) so the amount of heat available will be less.

That is a good point. If the PIIS melts more slowly during these cold years, the amount of meltwater at PIB will be smaller compared with periods when PIIS melts faster.

However, a weakening of the PIB gyre can decrease the ocean circulation locally and reduce the export of meltwater from PIB, which would concentrate the meltwater in the region. Thus, a relatively warm signal will be entering TEIS cavity compared with sea-ice-free periods when the gyre strength is not reduced. To address the question of the reviewer, we included the following sentences in Lines 291-295: “*Our hypothesis implies, somewhat counter-intuitively, that these ice-shelf cavities could warm slightly more during cold ocean periods if that condition supports landfast sea-ice cover in PIB and adjacent to TEIS, by ultimately allowing more meltwater to enter the ice shelf cavities, compared with sea-ice-free periods (Fig. 2c). However, there will be feedbacks upstream that may alter the concentration of meltwater in adjacent ice shelves.*”